

# Non-sea-salt aerosols that contain trace bromine and iodine are widespread in the remote troposphere

Gregory P. Schill[1], Karl D. Froyd[2,3], Daniel M. Murphy[1], Christina J. Williamson[4,5], Charles A. Brock[1], Tomás Sherwen[6], Mat J. Evans[7,8], Eric A. Ray[1,3], Eric C. Apel[9], Rebecca S. Hornbrook[9], Alan J. Hills[9], Jeff Peischl[1,3], Thomas B. Ryerson[1], Chelsea R. Thompson[1], Ilann Bourgeois[10], Donald R. Blake[11], Joshua P. DiGangi[12], and Glenn S. Diskin[12]

[1]National Oceanic and Atmospheric Administration, Chemical Sciences Laboratory, Boulder, CO 80305, USA
[2]Air Innova Research and Consulting, Boulder, CO 80305, USA
[3]Cooperative Institute for Research in Environmental Sciences, University of Colorado, Boulder, CO 80305, USA
[4]Climate Research Programme, Finnish Meteorological Institute, 00101 Helsinki, Finland
[5]Institute for Atmospheric and Earth System Research/Physics, Faculty of Science, University of Helsinki, 00014 Helsinki, Finland
[6]Hephaestus Partners, Ealing, London, UK, W7
[7]National Centre for Atmospheric Science, University of York, York YO10 5DD, UK.
[8]Wolfson Atmospheric Chemistry Laboratories, Department of Chemistry, University of York, York YO10 5DD, UK.
[9]Atmospheric Chemistry Observations & Modeling Laboratory, NSF National Center for Atmospheric Research, Boulder, CO 80305, USA
[10]Université Savoie Mont Blanc, INRAE, CARRTEL, F-74200 Thonon-Les-Bains, France
[11]Department of Chemistry, University of California, Irvine, CA 92697, USA
[12]NASA Langley Research Center, Hampton, VA 23681, USA

**Correspondence:** Gregory P. Schill (gregory.schill@noaa.gov)

**Abstract.** Reactive halogens catalytically destroy $O_3$ and therefore affect (1) stratospheric $O_3$ depletion, and (2) the oxidative capacity of the troposphere. Reactive halogens also partition into the aerosol phase, but what governs halogen-aerosol partitioning is poorly constrained in models. In this work, we present global-scale measurements of non-sea-salt aerosol (nSSA) bromine and iodine taken during the NASA Atmospheric Tomography Mission (ATom). Using the Particle Analysis by Laser
5 Mass Spectrometry instrument, we found that bromine and iodine are present in 8-26% (interquartile range, IQR) and 12-44% (IQR) of accumulation-mode nSSA, respectively. Despite being commonly found in nSSA, the mass concentrations of bromine and iodine in nSSA were low, 0.11-0.57 pmol mol$^{-1}$ (IQR) and 0.04-0.24 pmol mol$^{-1}$ (IQR), respectively. In the troposphere, we find two distinct sources of bromine and iodine to nSSA: (1) a primary source from biomass burning, and (2) a pervasive secondary source. In the stratosphere, nSSA bromine and iodine mass increased with increasing $O_3$ concentrations; however,
10 higher concentrations of stratospheric nSSA bromine and iodine were found in organic-rich particles that originated in the troposphere. Finally, we compared our ATom nSSA iodine measurements to the global chemical transport model GEOS-Chem; nSSA bromine concentrations could not be compared because they were not tracked in the model. We found that the model compared well to our ATom nSSA iodine measurements in the background atmosphere, but not in the marine boundary layer, biomass burning plumes, or in the stratosphere.



## 1 Introduction

Reactive halogens, including the reactive forms of bromine and iodine, affect $O_3$ loss in both the troposphere and stratosphere, and therefore contribute to uncertainties in global radiative forcing and stratospheric $O_3$ loss, respectively. In the troposphere, incorporating halogen chemistry into global chemical transport models decreases global tropospheric $O_3$ burdens by $\sim$17-19%, with >70-95% of the global odd oxygen loss from halogens coming from bromine and iodine (Saiz-Lopez et al., 2014; Sherwen et al., 2016b). In the stratosphere, bromine and iodine destroy $O_3$ analogous to chlorine, but bromine is approximately 45 times more effective than chlorine at destroying stratospheric $O_3$ on a per-atom basis (Daniel et al., 1999); iodine is even more efficient than bromine on a per-atom basis ($\sim$10$\times$, Koenig et al., 2020), suggesting that even a few tenths of a pmol mol$^{-1}$ of reactive iodine in the lower stratosphere can be significant for mid-latitude, lower stratosphere $O_3$ loss (Solomon et al., 1994; Saiz-Lopez et al., 2015).

Gas-phase bromine and iodine compounds are sourced from both anthropogenic and natural emissions, with natural oceanic emissions being the largest contributors. Once photolyzed, these gas-phase compounds form atomic Br and I radicals, which can catalytically destroy $O_3$. Both biotic and abiotic processes in the ocean form volatile organohalogens, including $CH_3Br$ (reactive lifetime, $\tau \sim$0.8 years) and $CH_3I$ ($\tau \sim$7 days), which are emitted directly into the marine boundary layer (MBL). Additionally, inorganic iodine (HOI, $I_2$) formed from the reaction of $O_3$ with iodine compounds in the ocean surface has been shown to be the dominant contributor to tropospheric reactive iodine budgets (Carpenter et al., 2013). Since 1950, this inorganic iodine source has been increasing, which can be attributed to enhanced $O_3$ concentrations from anthropogenic activity (Legrand et al., 2018) and enhanced sub-ice biological activity due to Arctic sea ice thinning (Cuevas et al., 2018). In addition to $CH_3Br$, long-lived halons sourced primarily from fire extinguishers and fire retardants are well-known contributors to reactive bromine concentrations; furthermore, modelling studies have shown that very short lived bromine species ($\tau <$6 months) can also contribute significantly to reactive bromine concentrations, even in the stratosphere (Stachnik et al., 2013; Keber et al., 2020).

In addition to gas-phase chemistry, multi-phase chemistry on aerosol and clouds can be a source of reactive bromine and iodine. For example, sea-salt aerosol (SSA) contain bromine at their saltwater concentrations, and models suggest that debromination of SSA can be the largest source of reactive bromine to the troposphere (Zhu et al., 2019). Consequently, heterogeneous reactions on aerosols and clouds are needed to correctly model reactive bromine concentrations (Schmidt et al., 2016; Badia et al., 2019). Iodine is often enriched in SSA (Murphy et al., 1997), but it is unclear whether that is from iodine enrichment in the sea-surface microlayer (Dean et al., 1963), or from fast gas- or multi-phase iodine chemistry.

While multi-phase chemistry can be a source of reactive halogens to the gas-phase, gas-phase bromine and iodine can also react to form secondary aerosol in non-SSA (nSSA). For example, bromine radicals are known to react with volatile organic compounds, and can form low-volatility products that partition into the aerosol phase (Ofner et al., 2012; Badia et al., 2019). The formation of higher iodine oxides ($I_2O_X$, X = 2, 3, 4) from reactive iodine precursors partitions iodine to nSSA, and can even cause new particle formation at coastal sites (O'Dowd et al., 2002) and in the Arctic (Allan et al., 2015) when reactive iodine concentrations are sufficient. Global chemistry models estimate that 15.3% of iodine emissions are converted into aerosol (Sherwen et al., 2016c), but this is dependent on the fate of higher iodine oxides, which is poorly understood



(Saiz-Lopez et al., 2014; Sherwen et al., 2016b). More recently, studies at the CERN cloud chamber have also shown that
reactive iodine emissions can react to form iodic acid (Finkenzeller et al., 2023), which is of sufficiently low volatility to cause
new particle formation on its own.

Quantitative nSSA bromine measurements are sparse, and often difficult to separate from SSA; however, filter-based measurements suggest that fine-mode aerosol are enhanced in $Br^-$ concentrations relative to sea salt (Sturges and Barrie, 1988; Hara et al., 2002). Qualitative measurements from the Particle Analysis by Laser Mass Spectrometry (PALMS) instrument also
suggest that bromine is present in nSSA in the upper troposphere/lower stratosphere (UT/LS), in roughly equal amounts to iodine (Murphy and Thomson, 2000). nSSA iodine measurements are more prevalent than bromine measurements, but are still sparse, especially in the free troposphere. A compilation of near-surface measurements suggest that aerosol iodine can range from 0.07-24 ng m$^{-3}$ (Saiz-Lopez et al., 2012). High-latitude UT/LS aerosol iodine measurements are of a similar magnitude (Koenig et al., 2020). Combined, these measurements suggest that nSSA bromine and iodine may be pervasive in the lower
atmosphere, akin to their gas-phase, reactive halogen counterparts; however, more measurements are needed across a wider range of latitudes, longitudes, and altitudes.

In this work, we present global-scale measurements of bromine and iodine in nSSA using the NOAA PALMS instrument. PALMS is a single-particle mass spectrometer, which measures both the number fraction of halogen-containing aerosol, as well as the absolute mass of halogen in these aerosol. Absolute mass measurements were constrained by new laboratory
calibrations of PALMS and independently measured particle size distributions. Measurements were made primarily during the NASA Atmospheric Tomography Mission (ATom). ATom consisted of four flight circuits that flew southward over the Pacific Ocean basin and northward over the Atlantic Ocean basin. Each circuit consisted of a series of 8-10 hour flights, with each flight conducting 6-10 vertical profiles from ~0.15 to 12 km. Additional measurements were made during two flight campaigns over the continental US [Deep Convective Clouds and Chemistry (DC-3) and Studies of Emissions and Atmospheric Composition,
Clouds and Climate Coupling by Regional Surveys (SEAC⁴RS)], and another flight campaign in the Tropical Atlantic [Costa Rica Aura Validation Experiment (CR-AVE)].

In these measurements, trace levels of iodine and bromine were ubiquitous in remote-tropospheric nSSA, regardless of season. Our measurements suggest that nSSA bromine and iodine have two sources in the troposphere–a primary source from biomass burning and a pervasive secondary source. We also found that nSSA iodine and bromine concentrations increased in
the stratosphere, but higher concentrations of bromine and iodine were measured in particles that originated in the troposphere. Single-particle measurements suggest this is due to their higher organic mass fraction, implying that organics help bind iodine in nSSA, or that organics serve as a proxy for another physico-chemical attribute that helps partition halogens to nSSA. Finally, we compared our nSSA iodine observations to a global chemical transport model; nSSA bromine was not explicitly tracked in the model. Outside of the MBL, biomass burning plumes, and the stratosphere, the model did well reproducing nSSA iodine
mass vertical profiles in both shape and magnitude. Iodine aerosol in the model is formed from the production of higher iodine oxides and HI, which form in the model via multi-step reactions involving $I_X$ (I and IO radicals). Thus, while we do not believe that our model-measurement comparison implicate higher iodine oxides and HI as the only iodine aerosol contributors during



ATom, they do suggest that I$_X$ is involved in the widespread, trace nSSA iodine concentrations we observed in the remote atmosphere.

## 2 Experimental

### 2.1 PALMS

The PALMS instrument configuration during the NASA ATom mission has been described in detail in several publications (*e.g.*, Schill et al., 2020). Briefly, PALMS operated inside the cabin of the NASA DC-8. Aerosol from outside the plane was sampled from the University of Hawaii's (UH) shrouded, near-isokinetic inlet (McNaughton, 2007), which has been maintained by the NASA Langley Aerosol Research Group Experiment (LARGE). The inlet was mounted to two window plates on the starboard side of the DC-8. Aerosol transmission through the inlet is essentially unity at small sizes. The transmission of large particles through the UH-LARGE inlet is pressure-dependent. The large size cut off (D$_{50}$) at low altitudes was measured to be 5.0 $\mu$m aerodynamic diameter; theoretical losses at 12 km have been calculated from these near-surface measurements, and the D$_{50,aerodynamic}$ is expected to be 3.2 $\mu$m. During ATom, PALMS detected particles between approximately 100 nm and 5 $\mu$m in geometric diameter, but the vast majority (99%) were 152 nm to 2.3 $\mu$m in geometric diameter. The lower limit for size detection in PALMS was set by noise in the optical system, which varied by flight due to differences in instrumental conditions. The upper size limit was mainly set by transmission through the pressure dependent, UH-LARGE aircraft inlet, the tubing from the aircraft inlet to the PALMS instrument, and the PALMS aerosol focusing lens.

An exemplary detection efficiency curve for PALMS, calculated for ATom-1 is presented in Fig. 6 in Froyd et al. (2019); however, in this work, we calculated number fraction and mass concentrations by coupling PALMS to an independently measured, quantitative size distribution (Froyd et al., 2019; Brock et al., 2021). For the size distribution measurements, we used the Aerosol Microphysical Package (AMP, Brock et al. 2019). AMP sampled air from the same UH-LARGE inlet, and downstream losses in tubing bends and due to gravitational settling were calculated and the data was corrected based on loss calculations (Brock et al., 2019). Using this coupling method, the PALMS detection efficiencies do not matter as long as a sufficient number of particles within each size bin are sampled and that PALMS measures each particle type within each size bin with similar detection efficiencies. To couple PALMS to the size distribution measurements, single-particle mass spectra were first classified into broad particle types such as biomass burning, sulfate-organic-nitrate mixtures (without biomass burning material), mineral dust, sea salt, and others. We then mapped the PALMS particle type fractions to the size distribution measurements using 4 bins (bin start/stop: 100 nm, 252 nm, 504 nm, 1128 nm, and 5036 nm), which generated mass concentrations for each particle type. The coupled PALMS-AMP size distributions required at least 5 particles in each bin. An inherent assumption of this technique is that the particle type fractions are constant within each bin. Sub-components of each particle type, *e.g.* organics and sulfate, can also be calculated from laboratory calibrations. For all nSSA particle types, we calculated the organic and sulfate mass concentrations (Froyd et al., 2019); the relative statistical error associated with this technique are ~10-25%, for sulfate and organic mass concentrations, if those concentrations are >0.01 $\mu g\ m^{-3}$. These errors are comparable to the estimated error of



115    the size distributions (Brock et al., 2019). For this study, we generated nSSA bromine and iodine mass concentrations using
the same method (see Sect. 2.2).

Mass concentrations for particle types and for halogens/chemical constituents were computed at three-minute intervals,
which strikes a balance between time resolution and particle statistics. All curtain plots and vertical profiles were the average
or median of these three-minute values.

## 2.2 PALMS Bromine and Iodine Mass Calibration

To calibrate PALMS for bromine and iodine mass in nSSA, we spiked trace amounts of bromine and iodine in a nSSA proxy. In
this work, we define nSSA as non-refractory particles that PALMS classifies into four broad chemical types: sulfate-organic-
nitrate, biomass burning, vanadium-containing (largely sulfate and organic, and generally from sea-shipping activity), and
meteoric-sulfuric acid particles (stratospherically sourced). The nSSA proxy consisted of ammonium sulfate and a mixture of
125    equal parts adipic and succinic acid. The organic mass fraction ($\sim$0.6) and organic O:C ratio (0.8) were chosen to represent
campaign averages from the ATom mission. We performed separate calibrations for positive and negative spectra. For bromine
and iodine, ammonium bromide and ammonium iodide were used. Positive and negative calibration spectra are similar to
typical nSSA mass spectra from ATom (Fig. A1).

Trace amounts of bromine or iodine (0.00055-0.10 mass fraction) were calibrated to a single peak in a single particle's
130    positive or negative mass spectrum (Fig. A1). The area under m/z 79 (A79) was chosen for bromine, and the area under m/z
(A127) was chosen for iodine. Both A79 and A127 were modified to exclude spectra with confounding, non-halogen A79
or A127. For example, modified A79 (MA79) excluded spectra where A81 was less than 2x A79, which is consistent with
bromine isotopic ratios. Further, MA79 excluded negative and positive spectra where A79 was less than 5x A95 and 5x A97,
respectively, which excludes spectra with large A79 contributions from methanesulfonic acid (MSA) fragmentation. MA79
also excludes negative spectra where A79 is less than 5x A63, which excludes particles that contain phosphorous ($PO_3^-$).
Finally, MA79 excludes spectra where A79 is less than 5x A77+A78, which excludes spectra that contain a large number of
organic peaks out to m/z 79. Modified A127 excluded positive and negative spectra where A129 was larger than A127. Like
MA79, this filtered out rare spectra with a large number of organic peaks out to m/z 127.

A calibration curve was made by taking the average of MA79 or MA127 at a given mass fraction and fitting the natural
log of those averages to a fourth-order polynomial (Fig. A2). For both MA79 and MA127, a separate curve was made for
positive and negative spectra. We did not extrapolate below the lowest values in each calibration curve; instead those values
were automatically set to zero. We did, however, add an artificial point at bromine/iodine mass fraction equals 1 (MA79 or
MA127 = 1) to ensure that the polynomial fit behaves well at higher mass fractions. The highest mass fractions in the bromine
and iodine calibrations were 0.077 and 0.044, respectively. During ATom, 99.98% of nSSA contained <0.077 bromine by mass,
and 99.97% of nSSA contained <0.044 by iodine by mass. Furthermore, the interquartile range of nSSA mass fractions during
ATom was 0.0012-0.0049 for bromine and 0.0011-0.0036 for iodine; thus, the vast majority of the observations are within the
calibration range.



For a given mass fraction of bromine or iodine, higher ion signal from a particle correspond to the higher probabilities that enough bromine or iodine ions arrive at the microchannel plate (MCP) to compete with instrumental noise. Thus, we used two
calibration curves each for both positive and negative spectra: one curve was made from calibration mass spectra with high total ion signal (high MCP output), and one for low total ion signal (low MCP output, Fig. A2). The low/high MCP output cutoffs for positive and negative spectra are $1 \times 10^{-10}$ C and $5 \times 10^{-11}$ C, respectively. These cutoffs were chosen from histograms of single-particle nSSA MCP outputs during all four AToms.

In order to couple PALMS data to an independently measured optical particle counter (Froyd et al., 2019), we took a three-
minute average of MA79 or MA127 prior to applying the calibration. MCP output for each particle were averaged in that three-minute period to determine if the low or high calibration curve was used. The calibration normalized root mean square error (RMSE) shrinks with higher bromine or iodine mass fractions and/or as more particles are collected (Fig. S3). Fortunately, it takes as few as 5-10 single particles for the normalized RMSE to plateau; thus, having >10 particles in a three-minute period only marginally reduces the error of these measurements. Both positive and negative mass spectra were used for bromine and
iodine mass calculations. If a given three-minute period has both positive and negative mass spectra, a time-weighted average was calculated.

Despite the similarity between the calibration spectra and typical bromine- and iodine-containing nSSA mass spectra, bromide and iodide are not the only forms of nSSA bromine and iodine. For iodine aerosol, the main species are iodide, iodate, and organo-iodine (Saiz-Lopez et al., 2012), but their ratios are not universal. While we did not use a proxy for organo-iodines,
similar calibrations were done with ammonium iodate, and yielded the same relationships as those shown in Fig. A2. The majority of nSSA bromine is expected to be in soluble bromide form (Sander et al., 2003). Nonetheless, the nSSA bromine and iodine measured in this study were often found at latitudes, longitudes, and altitudes where detailed nSSA halogen speciation has not been determined. Because we do not know the speciation of the ATom nSSA bromine and iodine, we also do not know their ionization efficiencies relative to our laboratory-generated nSSA with bromide and iodide. Thus, the measurements re-
ported here are susceptible to systematic errors. A factor of two would be a generous estimate of these systematic errors. There are several lines of evidence for this: (1) bromide and iodide + iodate likely make up more than 50% of the nSSA bromine and iodine that we measured during ATom (Gómez Martín et al., 2022), (2) the nSSA bromine and iodine in the ATom and calibration mass spectra were predominately attributable to only the atomic bromine and atomic iodine peaks, and (3) a comparison of PALMS nSSA iodine mass to AMS iodine mass measurements during ATom-1 and -2 from Koenig et al. (2020) show a
median bias of only +8% for PALMS. The $25^{th}$ and $75^{th}$ percentile biases are within a factor of two from the median bias. The AMS and PALMS measurements were co-located, the AMS transmission efficiency curve was applied to the PALMS data, and the data were binned by latitude (2 degree bins) and altitude (0.5 km bins). The AMS measurements were calibrated to three different iodine species: iodide, iodate, and an organo-iodine proxy (Koenig et al., 2020).

The total ion signal dependence was built into the bromine and iodine mass calibrations; thus, a significant correlation
between MA79 or MA127 and MCP output does not exist (Fig. A4); however, when calculating number fractions (i.e., the number of nSSA in an air mass with MA79 or MA127 > 0), a positive relationship between the bromine or iodine number fraction and MCP output does exist. Thus, our calculated number fractions are likely a lower limit to the true number fractions



of bromine- or iodine-containing nSSA. Considering this lower limit, we only used negative mass spectra for bromine number fraction calculations (*i.e.*, MA79 > 0); likewise, we only used positive mass spectra for iodine number fraction calculations (*i.e.*, MA127 > 0). In these two scenarios, using only the most sensitive mass spectrum polarity calculates our highest estimates of bromine- or iodine-containing nSSA without gratuitous filtering.

## 2.3   Regional and Altitude Boundary Definitions

To separate vertical profiles into tropics, mid-latitude, and polar regions, we used the same latitude cuts found in Table S1 in Schill et al. (2020). Briefly, the latitude cuts were based on changes in the zonal winds as well as concentration gradients of gas-phase and aerosol products. We separated the UT from the LS by using a combination of altitude, $O_3$, and CO concentrations. Specifically, we defined the stratosphere as air that is above 8 km and has $[O_3]$ / $[CO]$ > 3. Finally, the identification of the top of the marine boundary layer (MBL) is described in Brock et al. (2021). Briefly, the top of the MBL is defined by an abrupt change in temperature, dew point, wind speed / direction, and gas-phase tracers such as $O_3$, $NO_2$, and CO. Each MBL top was inspected by hand as oftentimes the location of the the MBL top was ambiguous, especially in regions where the MBL was not uniformly mixed.

## 2.4   Auxiliary Gas-Phase Measurements

In this work, we used $O_3$, HCN, $CH_3CN$, and $H_2O$ gas-phase measurements to support the PALMS aerosol measurements. $O_3$ was used as a proxy for the depth into the stratosphere. During ATom, $O_3$ was measured with the NOAA $NO_yO_3$ chemiluminescence instrument (Bourgeois et al., 2020). During CR-AVE, $O_3$ was measured using the NOAA $O_3$ Photometer (Gao et al., 2008).

HCN and $CH_3CN$ were used as a qualitative measure for both biomass burning influence and the age of biomass burning influenced air. In situ HCN and $CH_3CN$ were measured from the NASA DC-8 using the NSF NCAR Trace Organic Gas Analyzer (TOGA) during ATom-1 to -4. The TOGA is an online fast gas chromatograph with quadrupole mass spectrometer (GC/MS) that sampled and analyzed for 35 seconds every 2 minutes during ATom flights (Apel et al., 2015). The TOGA was calibrated in flight for $CH_3CN$ using several different ppb- and ppm-level VOC mixtures including $CH_3CN$ that are dynamically diluted with zero air from an integrated clean air generator (CAG) to ambient mixing ratio levels. TOGA was calibrated for both HCN and $CH_3CN$ before and after each ATom deployment in the laboratory using ppb- and ppm-level standard mixes diluted to ambient mixing ratios using the CAG. HCN and $CH_3CN$ during DC-3 and SEAC[4]RS were measured with the University of Innsbruck proton-transfer-reaction mass spectrometer (PTR-MS, Wisthaler et al., 2002)

Water vapor was measured using the NASA diode laser hygrometer (DLH) instrument (Diskin et al., 2002).DLH measures water vapor mixing ratio along an external path using near-infrared diode laser absorption, providing a fast, high-precision measurement relatively free of sampling artifacts. Relative humidities with respect to liquid water and ice are calculated from mixing ratio using the static temperature and pressure data. Water vapor was used to qualitatively assess the potential effects of $RH_{water}$ on aerosol phase, water available in the gas phase, and the age of air (see 3.1).





The CH$_3$I measurements aboard the NASA DC-8 during ATom were measured with the University of California, Irvine Whole Air Sample (WAS, Colman et al., 2001). The WAS CH$_3$I measurements were used to validate the GEOS-Chem CH$_3$I measurements, as a rough estimation of the model's skill in reproducing reactive iodine precursors.

## 2.5 Back Trajectory Calculations

Back trajectories were used to estimate the number of days since an air mass sampled during ATom had encountered a fire
(Schill et al., 2020). The data product "days since most recent fire influence" was an average of 245 back trajectories. The cluster of 245 back trajectories was initialized every minute along the ATom flight tracks and was projected back 30 days using the TRAJ3D model (Bowman, 1993; Bowman and Carrie, 2002). The model used the National Center for Environmental Prediction's meteorological fields at high resolution ($0.5° \times 0.5°$).

Fire locations were taken from the MODIS FRP (Collection 6), Visible Infrared Imaging Radiometer Suite (VIIRS) 375-m
FRP, and GFED v2. Fire plume injection heights were estimated from MODIS and VIIRS, based on FRP (Sofiev et al., 2012). GFED2, which does not report FRP, was assumed to inject smoke into a well-mixed boundary layer. A back trajectory was considered to "cross a fire" if it coincided with the latitude and longitude of a MODIS, VIIRS, or GFED2 fire, and was also below the calculated or assumed fire height.

## 3 Results and Discussion

## 3.1 Geographic and Vertical Distribution of Non-Sea-Salt Aerosol Bromine and Iodine

Bromine and iodine were commonly found in nSSA during ATom, regardless of season (Fig. 1 and Fig. 2). The median number fraction (and interquartile range, IQR) of nSSA containing bromine and iodine was 18% (8-26%) and 25% (12-44%), respectively. The trace amounts of bromine and iodine means that they can easily fall below the detection limit (<0.00055 mass fraction) when the total numbers of ions from a particle is small because, for example, a particle was near the edge of
the ionizing laser beam. While this was accounted for in the iodine and bromine mass calibrations, it causes a low bias in their number fractions. If we only use those particles whose MCP output was in the top $50^{th}$ percentile, the number fraction IQR shifts from 8-26% to 18-45% for bromine, and from 12-44% to 33-72% for iodine. Such high number fractions, especially in the free troposphere, are indicative of a pervasive secondary source of nSSA bromine and iodine whereby gas-phase bromine and iodine condense onto particles.
The pervasive nature of nSSA iodine and bromine is highlighted by a general lack of geographic and vertical trends. For example, the north-south, Atlantic-Pacific, and vertical gradients of nSSA bromine and iodine during ATom were weak, unlike many primary and secondary aerosol species. There were, however, some geographical highlights. The tropical Atlantic UT, which was typically deficient in primary aerosol, was especially rich in halogen-containing nSSA. For example, during ATom-2, the number fraction of nSSA containing iodine in the tropical Atlantic UT was generally over 80%. In contrast, nSSA in the
remote MBL and remote tropical Pacific were typically deficient in nSSA bromine and iodine. The MBL and remote tropical



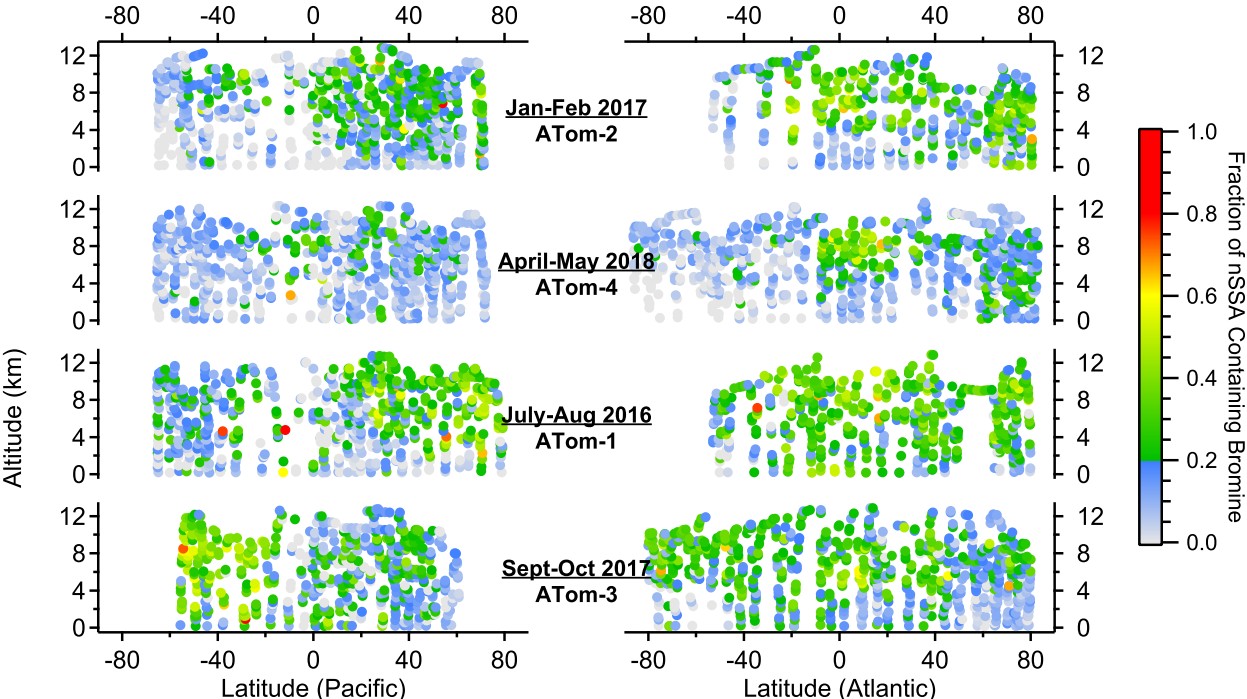

**Figure 1.** Altitude-latitude curtain plots of the number fraction of nSSA that contain bromine (*i.e.*, MA79 > 0) from ATom-1 to -4. The color bar was chosen to visually delineate when more than 20% of nSSA in an air mass contain bromine. Trace amounts of bromine in particles can lead to under counting in particles with low total ion signal (see Section 3.1); thus, the number fractions presented in this figure are considered to be lower limits.

Pacific also contained the lowest $O_3$ concentrations measured during ATom (Bourgeois et al., 2020). We found that the number fraction of nSSA that contain bromine and/or iodine monotonically increased with increasing tropospheric $O_3$ up to 50-60 ppb, where it levels out (Fig. A5 and Fig. A6). This suggests that, below 50-60 ppb, $O_3$ may be a limiting reactant in the pathways that convert reactive bromine or iodine into their aerosol-bound forms.

In addition to $O_3$, we also found that the number fraction of nSSA that contain bromine and iodine correlated monotonically with both temperature and $RH_{water}$ in tropospheric air (Fig. A5 and Fig. A6). We believe these correlations were the result of other processes. The correlation with temperature was weak (Spearman's rho: -0.42 to -0.47), and became insignificant below 273 K. The correlation with $RH_{water}$ was stronger (Spearman's Rho: -0.48 to -0.78), and only got stronger at $RH_{water}$ < 75%. The driest air sampled during ATom also corresponds to air that has been descending; thus, drier air contains more aged

aerosol, allowing more time for particles to gain a detectable amount of secondary iodine or bromine.

Although bromine and iodine were ubiquitous in nSSA, their mass concentrations were small relative to major aerosol constituents like organics and sulfate (Fig. 3 and Fig. 4). Median concentrations of bromine and iodine in nSSA were 0.7



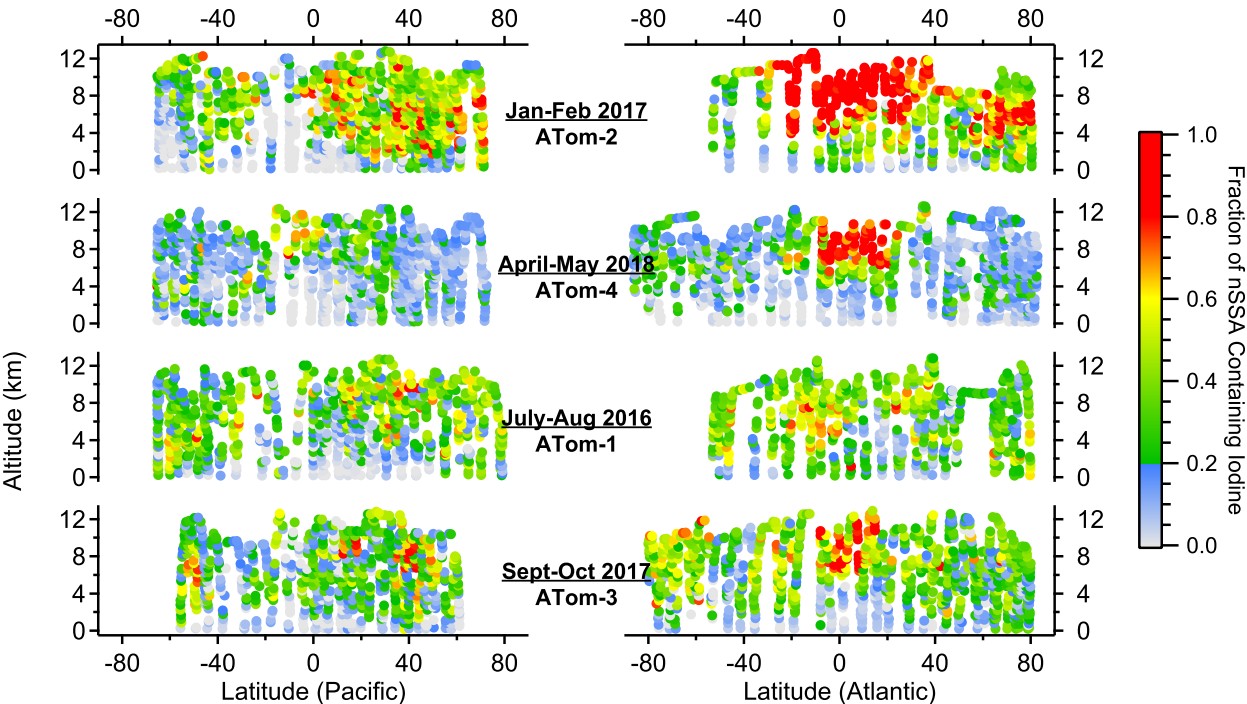

**Figure 2.** Altitude-latitude curtain plots of the number fraction of nSSA that contain iodine (*i.e.*, MA127 > 0) from ATom-1 to -4. The color bar was chosen to visually delineate when more than 20% of nSSA in an air mass contain iodine. Trace amounts of iodine in particles can lead to under counting in particles with low total ion signal (see Section 3.1); thus, the number fractions presented in this figure are considered to be lower limits.

(IQR: 0.3-1.5) and 0.4 (IQR: 0.16-1.0) ng m$^{-3}$, respectively. This corresponds to 0.25 (0.11-0.57) pmol mol$^{-1}$ and 0.10 (0.04-0.24) pmol mol$^{-1}$, respectively. During ATom, the median concentrations of organic and sulfate aerosol measured by PALMS were ~100 ng m$^{-3}$. Thus, while halogens were ubiquitous in nSSA during ATom, they were only ubiquitous at trace mass concentrations.

Large nSSA iodine and bromine concentrations (>1 pmol mol$^{-1}$) were consistently found in the tropical Atlantic at altitudes below ~6 km (Fig. 3 and Fig. 4). In these areas, we also encountered biomass burning plumes from the African continent, where biomass burning particles were often >80% of the accumulation-mode aerosol number, and biomass burning aerosol mass concentrations were ~1 μg m$^{-3}$ (Schill et al., 2020). In Section 3.2.1, we explore biomass burning as a source of primary nSSA bromine and iodine mass.

Elevated nSSA bromine and iodine concentrations (~0.3-0.4 pmol mol$^{-1}$) were also found in the lowermost stratosphere (defined as air masses where the altitude was > 8 km and the ratio of [O$_3$] to [CO] was > 3). Because of the ~12-13 km ceiling of the DC-8, the lowermost stratosphere was sampled during ATom only at polar latitudes and in mid-latitude tropopause folds



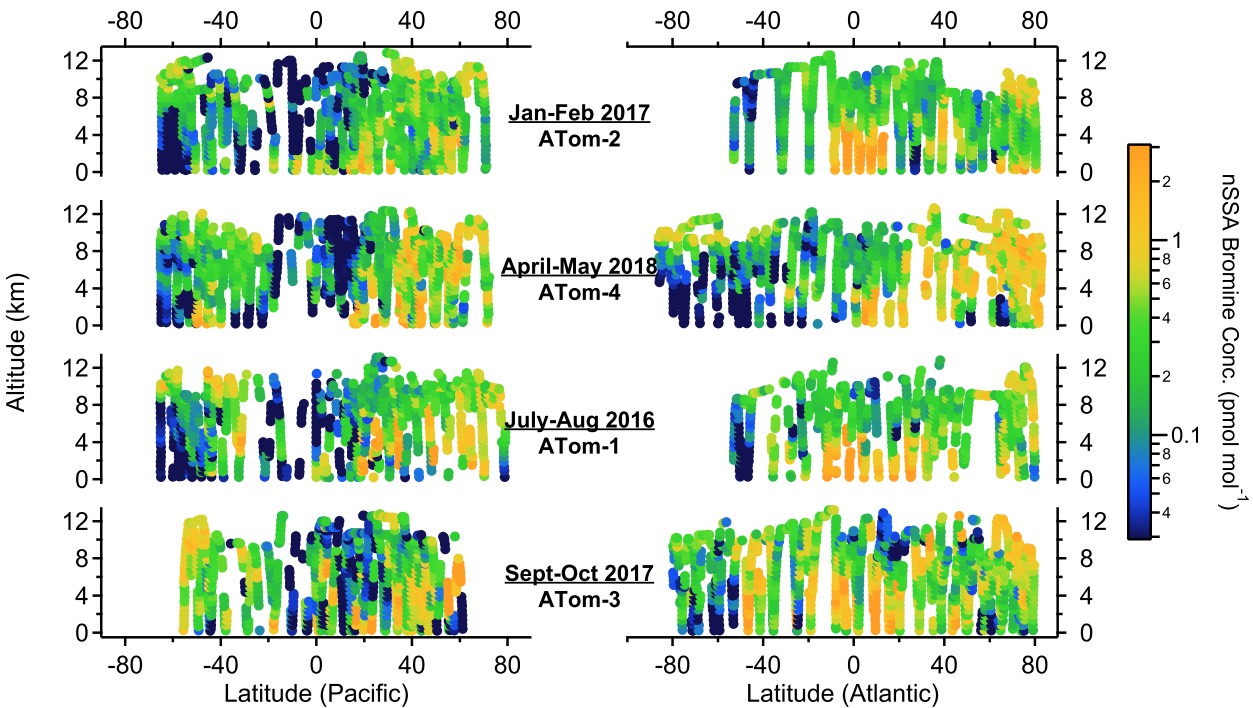

**Figure 3.** Altitude-latitude curtain plots of nSSA bromine mass concentrations (pmol mol$^{-1}$) from ATom-1 to -4.

(Murphy et al., 2021). Elevated nSSA iodine concentrations in the lower stratosphere are consistent with the results from Koenig et al., (2020), who measured an increase in aerosol iodine concentrations when transitioning from the UT to the LS using an aerosol mass spectrometer. These elevated concentrations suggest that the stratosphere may be a source of nSSA bromine and iodine; however, we found that most of the nSSA bromine and iodine in the lowermost stratosphere are found on tropospherically sourced particles. We explore this further in Section 3.2.3.

Finally, the vertical profiles of bromine and iodine mass in nSSA were similar in both their shapes and magnitudes (Fig. 5). Similar vertical profile shapes suggests that there are similar sources and sinks of nSSA bromine and iodine. In many cases, both bromine and iodine were found in the same particle (Fig. A1). Of all the negative spectra sampled during ATom, 47% of nSSA that contained bromine also contained detectable amounts of iodine; similarly, 59% of nSSA that contained iodine also contained bromine. Similar bromine and iodine masses in nSSA suggest that a higher fraction of reactive iodine partitions

into the aerosol phase than reactive bromine. Estimates of total reactive bromine (Br$_y$) in the global troposphere are ∼3 pmol mol$^{-1}$ (Sherwen et al., 2016b). Conversely, estimates from the same model suggest that total reactive iodine (I$_y$) in the free troposphere is only ∼0.5 pmol mol$^{-1}$. Thus, in the global troposphere, nSSA bromine is only ∼8% of the total reactive bromine concentration, whereas nSSA iodine makes up ∼20% of the total reactive iodine concentration.





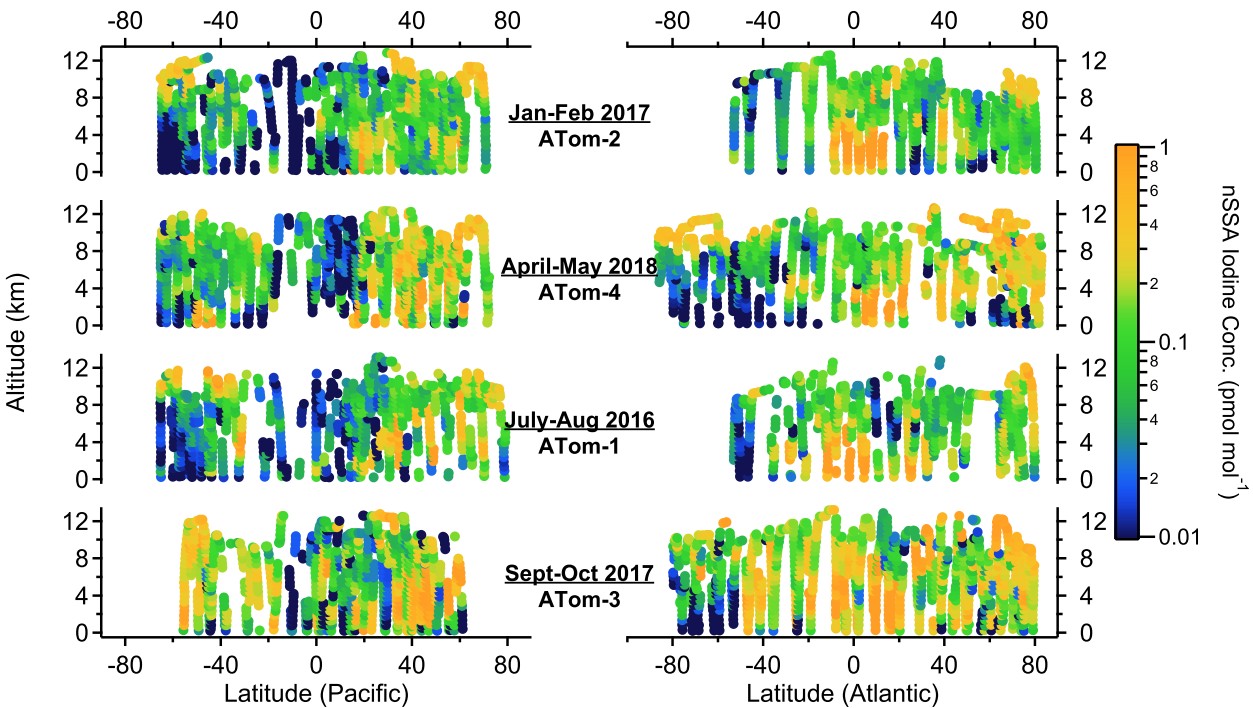

**Figure 4.** Altitude-latitude curtain plots of nSSA iodine mass concentrations (pmol mol$^{-1}$) from ATom-1 to -4.

## 3.2 Sources of Bromine and Iodine in Non-Sea-Salt Aerosol

From both the curtain plots and vertical profiles of nSSA bromine and iodine number fractions and mass, we suggest that there are two sources of bromine and iodine to tropospheric nSSA: (1) a primary biomass burning source, and (2) a pervasive secondary source from reactive uptake of gas-phase species. Additionally, we found that nSSA bromine and iodine mass concentrations increased in the lower stratosphere. Further classification of nSSA into stratospherically sourced and tropospherically sourced particles suggest that most of the stratospheric nSSA bromine and iodine were found on the tropospherically 290 sourced particles, and that organics may aid in binding bromine and iodine in nSSA. In the following sections, we will explore these sources in detail.

### 3.2.1 Biomass Burning

One advantage of using single-particle mass spectrometry instead of a bulk measurement is that we can separate nSSA halogen mass contributions from different aerosol types (*e.g.,* biomass burning vs. sulfate-organic-nitrate). While biomass burning and 295 sulfate-organic-nitrate mixtures are similar in their bulk composition (*i.e.,* they are both largely organics and sulfate by mass), biomass burning particles are distinct due to their potassium content (Schill et al., 2020). From the shape of each aerosol type's





vertical profile, we can surmise the sources and sinks of nSSA iodine and bromine. For example, the concentration of nSSA iodine and bromine mass at low-to-mid-altitudes was large in the tropical Atlantic. Separating nSSA iodine and bromine mass contributions from different particle types confirms that these high mass concentrations at 0-4 km in the tropical Atlantic were

due to iodine and bromine in biomass burning aerosol (Fig. 6 and Fig. A7). Furthermore, as these biomass burning plumes were convectively lofted and underwent wet removal, biomass burning aerosol mass decreased, as does the associated iodine and bromine mass.

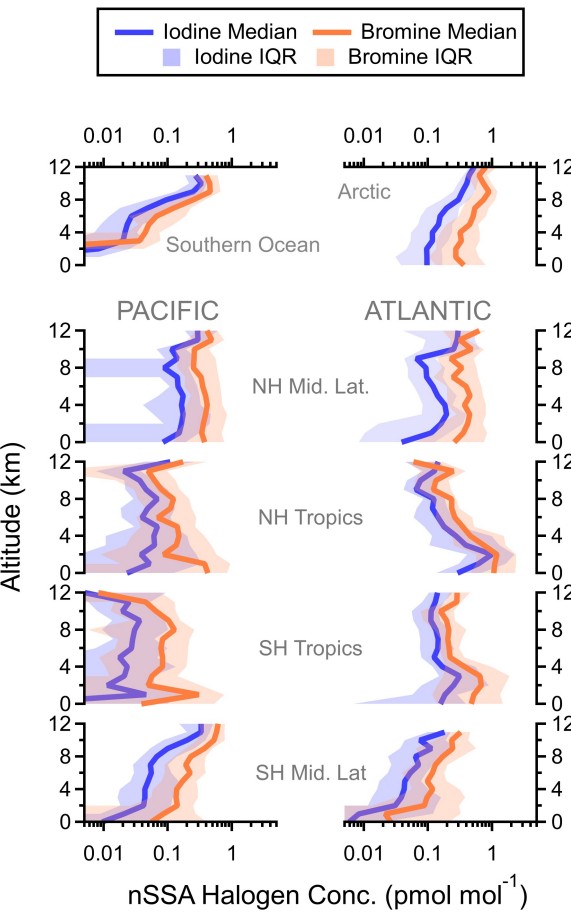

**Figure 5.** Regional vertical profiles of nSSA halogen mass (pmol mol$^{-1}$) from all four AToms. Latitude cuts for each region are in Table S1 in Schill et al. (2020).

Although the biomass-burning-associated iodine and bromine aerosol mass was high in these plumes (global max ~1 pmol mol$^{-1}$), the mass fraction of iodine and bromine in individual biomass burning particles was low, ~0.3% (Fig. A8). Thus,

fresh biomass burning particles have a small amount of iodine and bromine per particle, but high biomass burning aerosol mass concentrations in a plume amount to large absolute iodine and bromine masses.





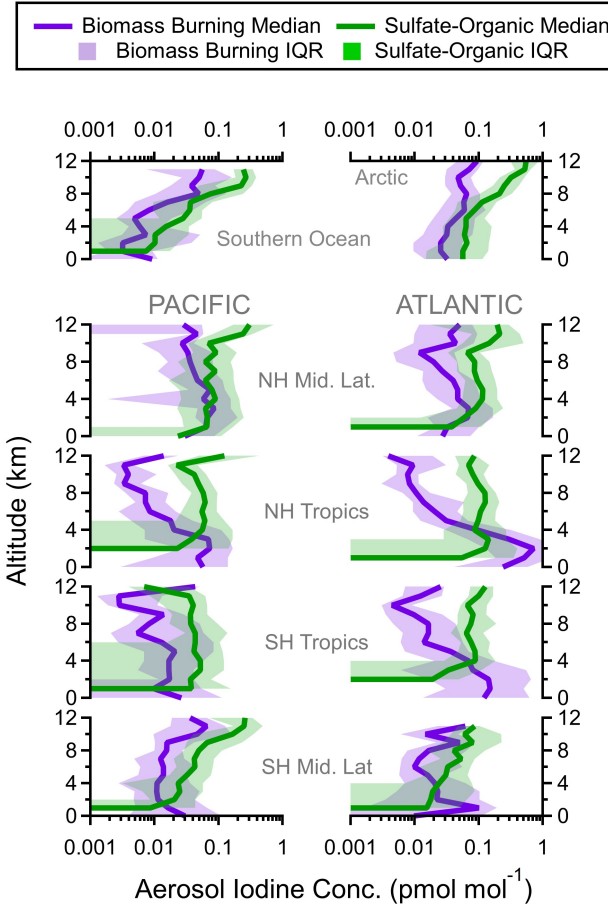

**Figure 6.** Regional vertical profiles of iodine mass (pmol mol$^{-1}$) in biomass burning and sulfate-organic-nitrate particles. Median values (solid lines) and interquartile ranges (shading) are from all four AToms. Latitude cuts for each region are in Table S1 in Schill et al. (2020).

Plotting nSSA iodine and bromine mass against several biomass burning tracers measured during ATom (Fig. 7 and Fig. A9) confirms a primary source of nSSA iodine and bromine from biomass burning. To extend these results to higher biomass burning tracer concentrations, we also include data from DC-3 (Barth et al., 2015) and SEAC$^4$RS (Toon et al., 2016). Several

large biomass burning events were encountered in both campaigns; however, all of the data from each campaign was used regardless of tracer concentrations. We find that there was a near-monotonic increase of iodine and bromine mass in nSSA with increasing concentrations of both CH$_3$CN and HCN (Fig. 7). From these plots, if we encountered at least 300 pptv of CH$_3$CN (*i.e.,* if we're in a biomass burning plume), then we generally found that median values of nSSA iodine and bromine mass were >1 pmol mol$^{-1}$; further, as gas-phase biomass burning tracers diluted and dissipated, so did nSSA iodine and bromine mass

concentrations, corroborating the vertical profile analyses above.





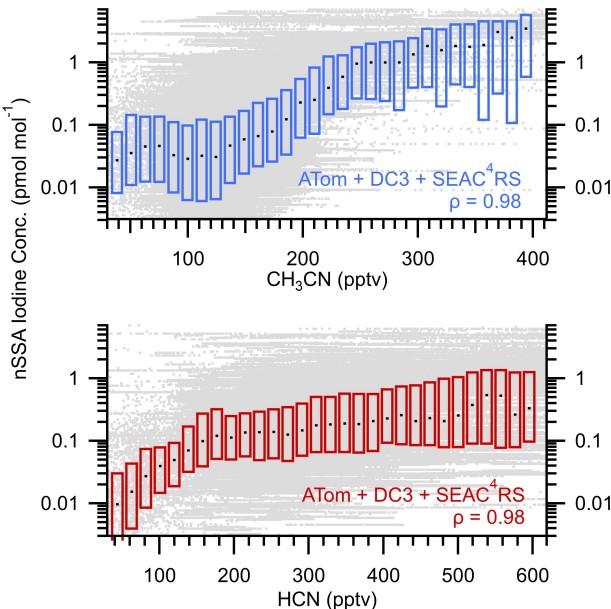

**Figure 7.** Box plots of nSSA iodine mass (pmol mol$^{-1}$) vs. gas-phase biomass burning tracers CH$_3$CN (pptv) and HCN (pptv). Raw data are gray dots, median values are black horizontal dashes, and the interquartile range is bound by the colored boxes. Median value Spearman's rho ($\rho$) for each is 0.98, which indicates near-monotonic relationship and suggests that biomass burning is a primary source of nSSA iodine mass.

The above discussion could imply that nSSA bromine and iodine are always emitted from biomass burning, with little source variation; however, the amount of nSSA bromine and iodine emitted from biomass burning is likely dependent on both fuel type and flaming conditions. This has been shown for methyl chloride and particulate chloride in laboratory burns of Douglas fir and duff beds (Reinhardt and Ward, 1995), where smoldering conditions favor methyl chloride emission and
flaming conditions promote chlorine formation in fine particulate matter. Similar studies for nSSA bromine and iodine, to the authors knowledge, do not exist. Furthermore, while there is guidance for fuel-type specific emissions of methyl halons (Andreae, 2019), nSSA bromine and iodine emission factors specific to fuel types and fire condition are largely unknown. Finally, fires that affect aerosol populations in the remote atmosphere, which are sometimes days or weeks old according to back trajectory calculations, are likely a mixture of different fires that have different fuels and flaming conditions. This will
cause a smearing effect, and give us the general picture shown in Fig. 7 and Fig. A9.

### 3.2.2 Pervasive Secondary Source

Above the MBL, vertical profiles of iodine and bromine mass within sulfate-organic-nitrate particles were constant with altitude in the tropics, and increased with altitude in the extratropical and polar regions (Fig. 6 and Fig. A7). Sulfate-organic-nitrate particles are a general class of particles that are generated from a variety of primary and secondary sources, but are not from





biomass burning or shipping oil combustion, and they do not contain stratospherically sourced meteoric material. The shape of these vertical profiles were similar to vertical profiles of the average number of days since a back trajectory ensemble encountered the boundary layer (*Days Since Boundary Layer Influence*), which suggests a pervasive secondary source of both iodine and bromine to nSSA in the free troposphere whose precursors are sourced from the surface. By plotting the mass fraction of iodine and bromine in individual sulfate-organic-nitrate aerosol vs Days Since Boundary Layer Influence, we
confirm that the iodine and bromine mass fractions monotonically increased in sulfate-organic-nitrate particles as their average time since last encountering boundary layer air increased (Fig. 8 and Fig. A10).

As discussed in Sect. 3.2.1, biomass burning aerosol were a primary source of nSSA iodine and bromine; however, despite a large absolute mass concentration of iodine and bromine, the freshest biomass burning particles had low halogen mass fractions (Fig. A8). As biomass burning plumes aged and diluted into the background atmosphere, they accumulated iodine
and bromine and their iodine and bromine mass fractions increased (Fig. 8 and Fig. A10). Along with the shapes of the sulfate-organic-nitrate aerosol iodine and bromine vertical profiles, this suggests two non-exclusive lines of evidence for a pervasive, secondary source of nSSA iodine or bromine: (1) iodine and bromine partition to nSSA and accumulate as the particles age over long time periods, and (2) aged air masses, which have undergone more aerosol removal, have lower absolute nSSA concentrations, and will accumulate more iodine and bromine per particle per time in a pervasive background of precursors.

In the MBL, iodine and bromine mass in sulfate-organic-nitrate particles was often below our limit-of-detection (<0.00055 mass fraction). The remote MBL, however, is a known source of reactive iodine and bromine precursors (*e.g.*, Carpenter et al. 2013). Previous measurements suggest that the highest concentrations of reactive iodine radicals are found in the MBL (Volkamer et al., 2015); furthermore, debromination from acidic SSA is known to occur in the MBL and may be one of the largest sources of reactive bromine (Sander et al., 2003). Thus, we expect high nSSA halogen concentrations in the MBL;
however, from our measurements, it seems that the emission/formation of reactive halogen precursors in the MBL are not the limiting factor to forming nSSA iodine and bromine in the MBL. One possibility is that low $O_3$ or high water vapor concentrations are preventing the formation of iodine or bromine aerosol or affecting their partitioning from the gas phase to the aerosol phase (Sect. 3.1). Another possibility is that wet removal in the humid/wet MBL is faster than the slow accumulation of secondary bromine and iodine.

### 3.2.3 Stratosphere

During ATom, the DC-8 sampled lower stratospheric air, but this was limited to the high-latitude flights due to the DC-8's ~12-km ceiling (Murphy et al., 2021). In these flights, at altitudes greater than 8 km, we found that iodine and bromine mass in nSSA monotonically increased with $O_3$ concentrations until it plateaued near 0.3-0.5 pmol mol$^{-1}$ (Fig. 9 and Fig. A11). If we use $O_3$ concentration at altitudes greater than 8 km as a proxy for penetration depth into the lower stratosphere, we see
that nSSA iodine and bromine increased from the UT to the LS. This is similar to the results found in Koenig et al. (2020), who showed an increase in nSSA aerosol iodine in the lower stratosphere was concomitant with a decrease in iodine oxide radical (IO) concentrations. The IO measurements by Koenig et al., however, were taken primarily in the tropical UT/LS. Thus, in addition to the high-latitude UT/LS data from ATom, we also include data from the CR-AVE mission, which sampled the




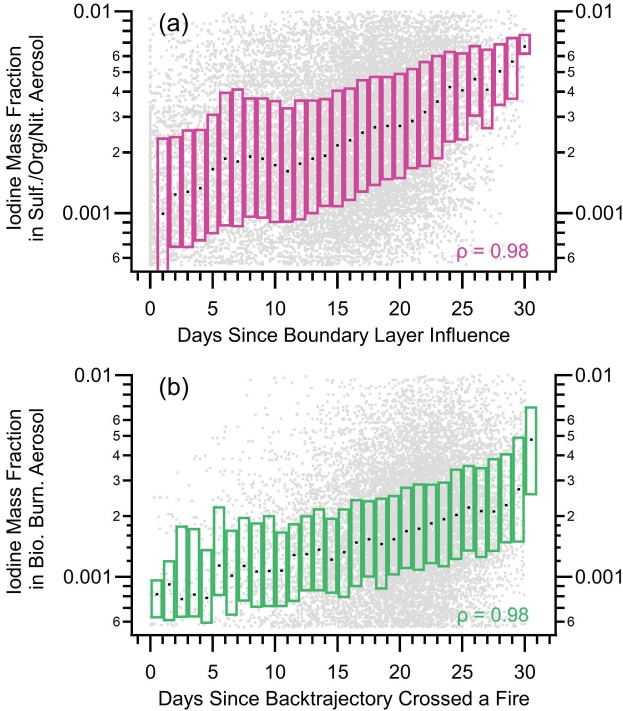

**Figure 8.** Box plots of nSSA iodine mass fraction vs. two back trajectory products, (a) the average number of days since a back trajectory was in the boundary layer, and (b) the number of days since a back trajectory crossed a fire. Raw data are gray dots, median values are black horizontal dashes, and the interquartile range is bound by the colored boxes. Median value Spearman's rho ($\rho$) for each is 0.98, which indicates near-monotonic relationship and suggests that biomass burning is a primary source of nSSA iodine mass.

UT/LS in the tropics. Above ~8 km in altitude and ~25 ppbv $O_3$, nSSA iodine and bromine concentrations increased from the
UT to the LS in both polar (ATom) and tropical (CR-AVE) regions. This indicates that the enhanced partitioning of halogens to aerosol in the stratosphere may be a global phenomenon, but further measurements at other locations in the UT/LS are needed.

Fig. 9 and Fig. A11 suggest that there is a stratospheric source of nSSA halogens; however, iodine and bromine were not evenly distributed among tropospheric and stratospheric particle types, which are easily differentiated by PALMS (Murphy et al., 2021). Calculating nSSA iodine and bromine mass fractions for different PALMS particle types reveals that most of
the nSSA iodine and bromine mass in the stratosphere resided on tropospherically sourced particles (Fig. 10 and Fig. A12). For example, stratospheric acidic sulfate particles with trace meteoric compounds (meteoric particles) sourced from the upper stratosphere contained ~0.1% iodine and ~0.5% bromine, while the tropospheric particles (biomass burning and tropospheric sulfate-organic-nitrate) contained 3-10x as much iodine and 1.5-2.5x as much bromine. In terms of absolute mass, there were more sulfate-organic-nitrate particles than meteoric particles in the LS (Fig. A13), so the lower mass fractions were not due to
the same amount of secondary iodine and bromine being spread out over more meteoric particles. Thus, while nSSA bromine



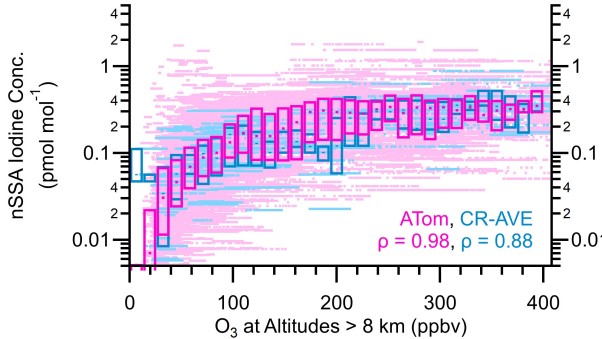

**Figure 9.** Box plots of nSSA iodine mass (pmol mol$^{-1}$) vs. O$_3$ for ATom (yellow) and CR-AVE (magenta). The median values of Spearman's rho ($\rho$) for ATom and CR-AVE are 0.98 and 0.88, respectively, which indicates near-monotonic relationship and suggests that the stratosphere may be a source of nSSA iodine mass globally.

and iodine increases in the LS due to a more efficient conversion of gas-phase reactive bromine and iodine to their aerosol-bound products (*e.g.*, Koenig et al. 2020), these products preferentially form on the tropospherically sourced particles.

Tropospherically sourced particles generally contain much more organic material than stratospherically sourced particles (Murphy et al., 2021), and previous PALMS studies have shown that mass spectra with higher iodine peaks were found in particles with higher organic content (Murphy et al., 1997). It has also been shown from aerosols collected at the Mace Head research station that over 90% of soluble aerosol iodine is organically bound, with the rest being iodide or iodate (Gilfedder et al., 2008). Additionally, HCl has been shown to be soluble in stratospheric wildfire particles with significant organic mass fractions, presumably organic acids (Solomon et al., 2023); HI may be similarly dissolved in these organics.

To explore the idea of organics retaining iodine in aerosol, we plotted iodine mass fraction against organic mass fractions for four different PALMS particle types in the lower stratosphere–two stratospherically sourced particles, meteoric and sulfuric acid, and two tropospherically sourced particles, biomass burning and sulfate-organic-nitrate (Fig. 11). As shown, higher mass fractions of iodine were found in particles with higher organic content. It should be noted that the organic mass fraction may be a proxy for or correlate to other physico-chemical properties (*e.g.,* particle neutrality), and that iodine preferably partitions into these organic-rich particles because of these proxied properties. The proxied properties may even be secondary effects; for example, more neutralized particles may affect organic acid gas-particle partitioning, which then may affect HI solubility in stratospheric particles. The importance of organics, either directly through organic binding / HI dissolution or as a proxy, likely extends into the troposphere, but it is difficult to unearth this relationship because of the two competing primary and secondary sources.

For bromine, there was a weaker correlation between organic mass fractions and nSSA-bromine concentrations in stratospheric particles (Fig. A14). This suggests that organic binding or HBr dissolution in small organic acids is not the only and perhaps not even the primary mechanism to retain nSSA bromine. Regardless, even though the relationship is less clear for bromine than it is for iodine, lower stratospheric particles with higher organic content did generally contain more bromine.





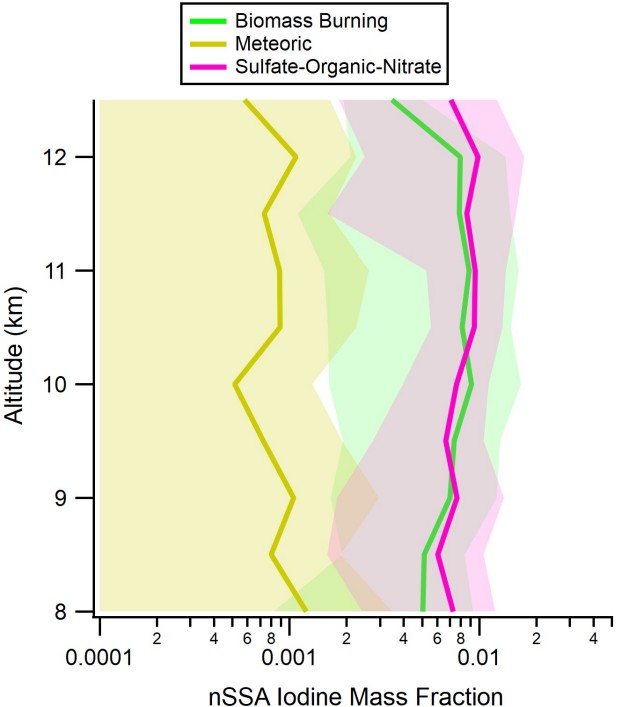

**Figure 10.** Vertical profiles of nSSA iodine mass fractions in the most common particle types found in the lower stratosphere during ATom: tropospherically sourced biomass burning and sulfate-organic-nitrate particles and stratospherically sourced meteoric-sulfuric-acid particles. Average values (solid lines) and standard deviations (shading) are from all four AToms. In this work, the lower stratosphere is defined as air masses above 8 km where the $[O_3]/[CO]$ ratio is greater than 3.

Thus, organics are also likely playing some role in retaining nSSA bromine or acting as a proxy for some physico-chemical properties that help retain nSSA bromine.

## 3.3 Iodine Modelling with GEOS-Chem

In this work, we compare the ATom measurements to the GEOS-Chem chemical transport model (www.geos-chem.org, version 12.9.1), which has an online NOx–VOC–HOx–Ox-BrOx-ClOx-IOx chemistry scheme (Sherwen et al., 2016b, 2017; Wang et al., 2019) computed for all 72 levels up to 0.01 hPa (Eastham et al., 2014). In GEOS-Chem 12.9.1, nSSA bromine is not tracked, but nSSA iodine is accounted for. The formation of iodine aerosol occurs through the uptake of gas-phase iodine on existing fine and coarse mode aerosol (Sherwen et al., 2016b, c), which is then transported and deposited following the model's treatment of its parent aerosol. In GEOS-Chem 12.9.1, iodine partitions into nSSA from the irreversible uptake of HI and higher iodine oxides ($I_2O_X$, X = 2, 3, 4). It can also partition into aqueous nSSA through several water soluble gas-phase iodine species, which either react to form new gas-phase species, or are released upon drying (see Table B5 in Sherwen et al., 2016b). The model was run at a horizontal resolution of 4x5° and transport was driven by offline meteorological fields from





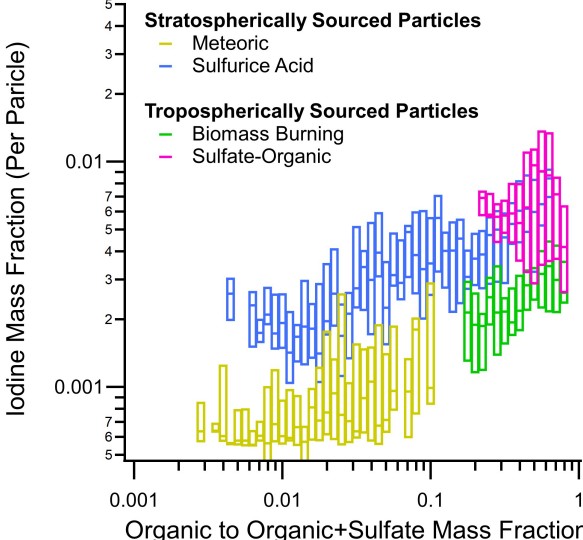

**Figure 11.** Box plots of iodine mass fraction in particles observed in the lower stratosphere as function of organic-to-organic-plus-sulfate mass ratio from all four AToms. Of the four dominant particle types, two are tropospherically sourced (biomass burning and sulfate-organic-nitrate) and two are stratospherically sourced (meteoric-sulfuric-acid and sulfuric-acid). Horizontal dashes are median values and the boxes are interquartile ranges. In this work, the lower stratosphere is defined as air masses above 8 km where the $[O_3]/[CO]$ ratio is greater than 3.

Goddard Earth Observing System Modern-Era Retrospective analysis for Research and Applications (MERRA-2 version 2; Gelaro et al., 2017). Global anthropogenic emissions were from the Community Emissions Data System (CEDS; Hoesly et al., 2018) and biomass burning emissions were from the Global Fire Emissions Database (GFED4; Mu et al., 2011). Inorganic emissions followed Carpenter et al., (2013), and organic emissions followed Ordóñez et al., (2012), The model was run for the entire period of the NASA ATom campaign, following a discarded year of "spin up," with data extracted for the nearest point

in space and time along the route taken by the aircraft.

Iodine aerosol formation in the model is missing many known processes [*e.g.,* $HIO_3$ formation (Finkenzeller et al., 2023), heterogeneous/multi-phase reactions (Saiz-Lopez et al., 2012), and/or halogen solvation by organics (Solomon et al., 2023)]; nonetheless, these model-measurement comparisons can help inform the model that their reactive iodine concentrations and the processes that transfer reactive iodine to the aerosol phase are roughly correct. A global surface evaluation of the model's

iodine aerosols is described in Sherwen et al., (2016c).

Vertical profiles of nSSA iodine mass from ATom measurements and GEOS-Chem 12.9.1 agree well in both magnitude and shape (Fig. 12). The agreement was best in the background free troposphere (Fig. A15)–*i.e.*, outside of air masses influenced by biomass burning aerosol ("influenced" is defined here as accumulation-mode biomass burning aerosol number fractions > 0.5), below the lower stratosphere, and outside of the MBL. Thus, the model seems to capture the pervasive secondary formation of

nSSA iodine well. In the model, higher iodine oxides are directly or eventually formed from self-reactions of IO. HI is formed



from the reaction of I and $HO_2$. This suggests that IO and I ($I_X$) are important precursors to the pervasive secondary nSSA iodine, and that the net flux of iodine to nSSA is similar to the reactive uptake of $I_2O_X$ with $\gamma = 0.02$ and HI with $\gamma = 0.1$.

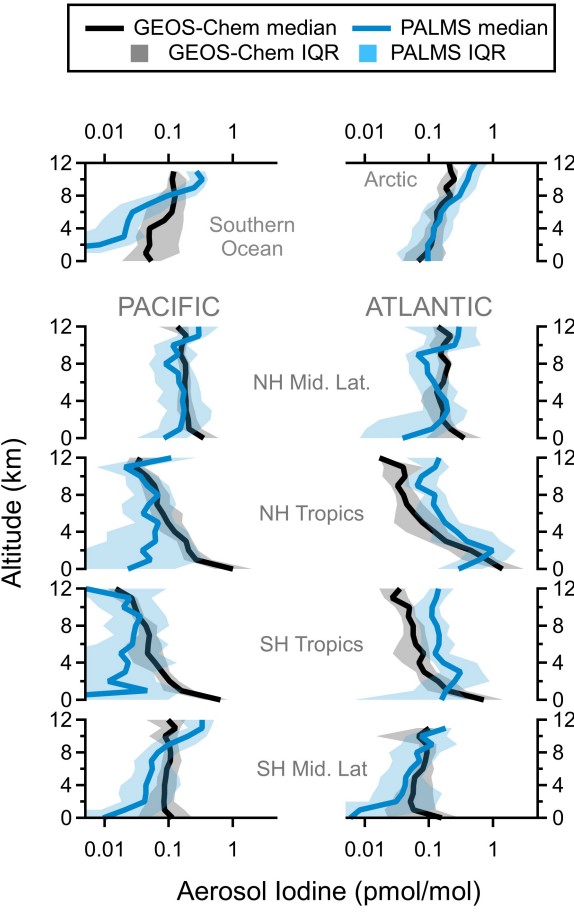

**Figure 12.** Regional vertical profiles of nSSA iodine mass (pmol mol$^{-1}$) from PALMS observations (blue) and GEOS-Chem 12.9.1 model output (black). Median values (solid lines) and interquartile ranges (shading) are from all four AToms. Latitude cuts for each region are in Table S1 in Schill et al. (2020).

The agreement between GEOS-Chem 12.9.1 output and the ATom measurements suggests that $I_X$ are important intermediates, or important tracers of the true intermediates, to nSSA iodine formation. The I radical is not routinely measured, and no
vertical profiles are known to the authors for comparison to the model. IO, however, has been measured from aircraft platforms (*e.g.*, Volkamer et al. 2015), and the I and IO concentrations are inexorably related as these species rapidly cycle between each other. The model does generally have a high bias to IO (Sherwen et al., 2016a); for example, a comparison to the TORERO aircraft IO measurements (Wang et al., 2015) shows that the model has a bias of $+83\%$ on average, with the greatest bias in the MBL ($+125\%$). This could explain some of the biases with the ATom measurements; unfortunately, a direct measurement
of IO was not made on the DC-8 during ATom.





The dominant source of $I_X$ in the atmosphere is the photolysis of primary inorganic iodine (*e.g.*, HOI, $I_2$) emitted from the ocean, although the photolysis of organic iodine species, primarily $CH_3I$, contributes in the free troposphere. During ATom, we also did not have direct measurements of HOI or $I_2$, but the model does perform well against ATom measurements of $CH_3I$ (Fig. A16). Thus, while nSSA iodine measurements suggest that $I_X$ is an important proxy for nSSA iodine precursors, more measurements of I, IO, $I_X O_Y$, HI, and other short-lived inorganic iodine species are needed to better constrain the mechanism of aerosol formation. These measurements, ideally, need to be accurate to the sub-0.1 pmol mol$^{-1}$ level in order to be a useful comparison to the nSSA iodine concentrations in this work.

As mentioned above, the model underestimated nSSA iodine in areas of high biomass burning aerosol influence, underestimated nSSA iodine in the stratosphere, and overestimated nSSA iodine in the MBL. Air masses strongly influenced by biomass burning aerosol, defined as air masses where biomass burning aerosol make up over half of the accumulation-mode aerosol by number, exhibited slight model underestimation, approximately $-66\%$ on average when giving each ATom ocean basin equal weight. This is because the model does not have an nSSA iodine source from biomass burning. Despite not having a biomass burning source of nSSA iodine, there were large variations in the nSSA iodine mean log bias and centered root mean squared error (CRMSE, Fig. A15). For example, in the ATom-2 Atlantic, the model underestimated nSSA iodine by an order of magnitude, and the spread of biases are also almost an order of magnitude. This variation suggests that different fuel types and flaming conditions may be important for forming iodine aerosol from biomass burning. Furthermore, in some air masses influenced by biomass burning, the mean log bias was near-zero; however, as shown in Fig. 12, this may be a convolution of underestimation of nSSA iodine in biomass burning plumes and overestimation of nSSA iodine in or just above the MBL.

Inside the MBL, the model overestimated nSSA iodine by a factor of 7, on average. In GEOS-Chem 12.9.1, aerosol iodine can condense on both nSSA and also fine-mode and coarse-mode sea salt. It is possible that the partitioning of iodine between nSSA and sea-salt aerosol is incorrect in the model. As mentioned above, the number fraction of iodine-containing aerosol was positively correlated with $O_3$ below 50-60 ppb. Thus, if the model was overestimating $O_3$ concentrations in the MBL, then both IO production and nSSA iodine may be overestimated; however, we find that, in the MBL, the model had a minimal low bias (approximately $-8\%$) compared to the ATom measurements of $O_3$. Water vapor has been shown to reduce the cluster formation of $I_X O_Y$ (Gómez Martín et al., 2020), which may prevent secondary iodine formation in the humid MBL. Additionally, in the humid and wet MBL, there may be increased wet deposition and rain out of both soluble gas-phase iodine species and hygroscopic aerosol that are underestimated in the model.

In the high-latitude LS, the model underestimated nSSA iodine mass by a factor of $\sim$2.4. Some portion of this missing nSSA iodine mass may be the contribution from biomass burning aerosol that were often present in the LS (*e.g.*, Fig. 6). In addition to this, other processes in the model may account for missing nSSA iodine, including incorrect photolysis rates of key gas-phase species like $I_X O_Y$, incorrect transport of very short-lived (VSL) iodine to the UT/LS, or increased uptake on cold, dry, and/or acidic particles. Additionally, iodine in the form of HI may be efficiently dissolved in the organic phase of stratospheric aerosol particles akin to HCl (Solomon et al., 2023). Missing chemistry in the model could also be driving model-measurement discrepancies in the stratosphere. Measurements from the CU HR-ToF-AMS during ATom show that aerosol iodine in the stratosphere was consistent with iodate, while most of the aerosol iodine found in the troposphere was iodide or organic iodine



(Koenig et al., 2020). In GEOS-Chem 12.9.1, nSSA-iodine is formed through the same mechanisms in the stratosphere as it is in the troposphere, which is inconsistent with the HR-ToF-AMS results. Additionally, PALMS measurements suggest that organics may be important for binding iodine in nSSA (Sect. 3.2.3). During the TORERO and CONTRAST campaigns, IO was reported to decrease from the tropical UT to the LS, suggesting that an additional source of iodine from the upper stratosphere is not needed to explain the increase in aerosol iodine (Koenig et al., 2020). Finally, the total available iodine may be limiting nSSA iodine concentrations in the LS, as nSSA iodine concentrations were similar between AToms despite highly variable aerosol environments (Murphy et al., 2021).

## 4   Conclusions

In this work, we have provided global-scale measurements of the accumulation-mode number fraction, mass fraction, and absolute mass of nSSA bromine and iodine. We find that these halogens were commonly found in free-tropospheric nSSA, and, as a lower estimate, bromine and iodine were present (*i.e.*, have mass fractions >0.00055) in 0.08-0.26 (IQR) and 0.12-0.44 (IQR) of the nSSA sampled during ATom, respectively. Despite being commonly found in nSSA, the mass concentration of these halogens was low, approximately 0.11-0.57 pmol mol$^{-1}$ (IQR), and 0.04-0.24 pmol mol$^{-1}$ (IQR) for bromine and iodine, respectively.

Using the capabilities of single-particle mass spectrometry, we have attributed several sources to the ubiquitous, trace bromine and iodine in nSSA. First, we find that there is a primary source of bromine and iodine in nSSA from biomass burning. Fresh biomass burning aerosol contain lower mass fractions of bromine and iodine than background particles, but higher aerosol mass concentrations in plumes suggest that >1 pmol mol$^{-1}$ of bromine and iodine can be found in nSSA in aged biomass burning plumes. The second source of bromine and iodine to nSSA is a pervasive secondary source. Here, organic and inorganic bromine and iodine sources from the oceans are converted into a pervasive background of reactive species that can form low volatility bromine and iodine products that partition into the aerosol phase. It is unclear what the ultimate fate of these aerosol-bound halogens are, but plotting mass fractions in individual particles as function of back trajectory age suggests that individual nSSA generally accumulate bromine and iodine as they age. Finally, we also find that there was an increase in bromine and iodine in stratospheric nSSA, but that these are concentrated on tropospherically sourced organic-sulfate mixtures, not the stratospherically sourced sulfuric acid particles that sometimes contain trace meteoric metals.

We have compared our results to output from the global chemical transport GEOS-Chem version 12.9.1, which has an online NOx–VOC–HOx–Ox-BrOx-ClOx-IOx chemistry scheme. Iodine aerosol is formed in the model, but there are no mechanisms for nSSA bromine to form. We find that the model compares well to our nSSA iodine measurements in the background troposphere, which corroborates that nSSA iodine is formed through secondary processes, likely involving I$_X$ (I and IO). While the agreement in the background atmosphere is good, the model underestimated nSSA iodine in biomass burning plumes and in the stratosphere. Additionally, the model overestimated the concentration of nSSA bromine and iodine in the marine boundary layer by a factor of 7. Thus, while the agreement outside of biomass burning plumes, the stratosphere, and the MBL suggest



that the iodine aerosol schemes are relevant to background aerosol, some sources and chemistry in the iodine aerosol schemes are missing.

505 *Data availability.* The data used in this paper are publicly available in online repositories. The ATom data can be found at https://daac.ornl.gov/ATOM/campaign/. The CR-AVE data can be found at https://espoarchive.nasa.gov/archive/browse/cr_ave/WB57. The DC3 data can be found at https://www-air.larc.nasa.gov/cgi-bin/ArcView/dc3. Finally, the SEAC[4]RS data can be found at https://www-air.larc.nasa.gov/cgi-bin/ArcView/seac4rs.

## Appendix A: Additional Figures

510 The following section includes figures that support the paper, but would disrupt the flow of descriptions or demonstrations. These figures support the PALMS bromine and iodine mass calibrations, show box plots that outline general trends of nSSA bromine and iodine with gas-phase and meteorological tracers, are the nSSA bromine companions figures to the nSSA iodine figures in the main text, or help us understand the biases and trends in the GEOS-Chem model.

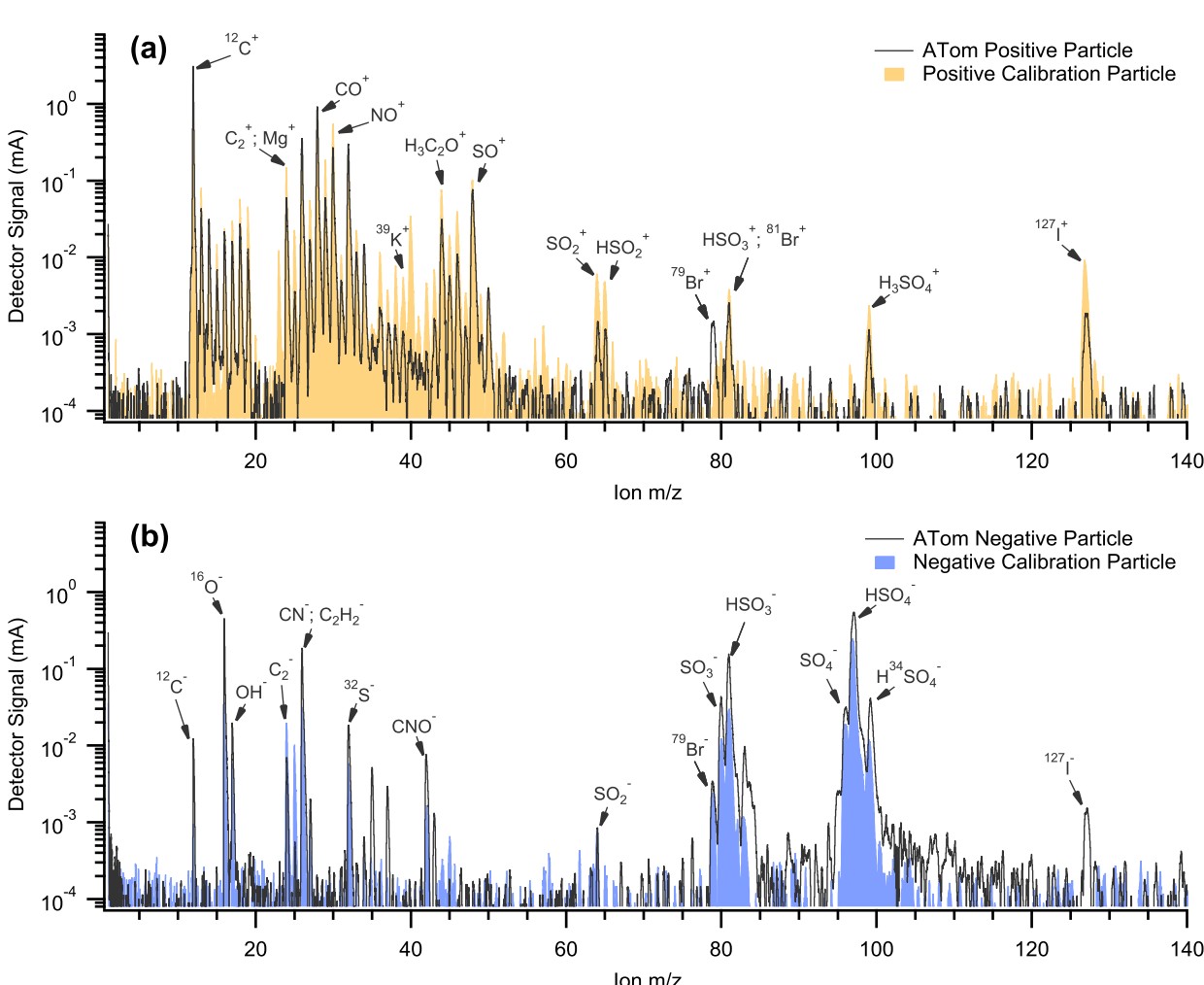

**Figure A1.** Example mass spectra of (a) a positive iodine calibration particle and an iodine-containing nSSA from ATom, and (b) a negative bromine calibration particle and a bromine-containing nSSA from ATom.



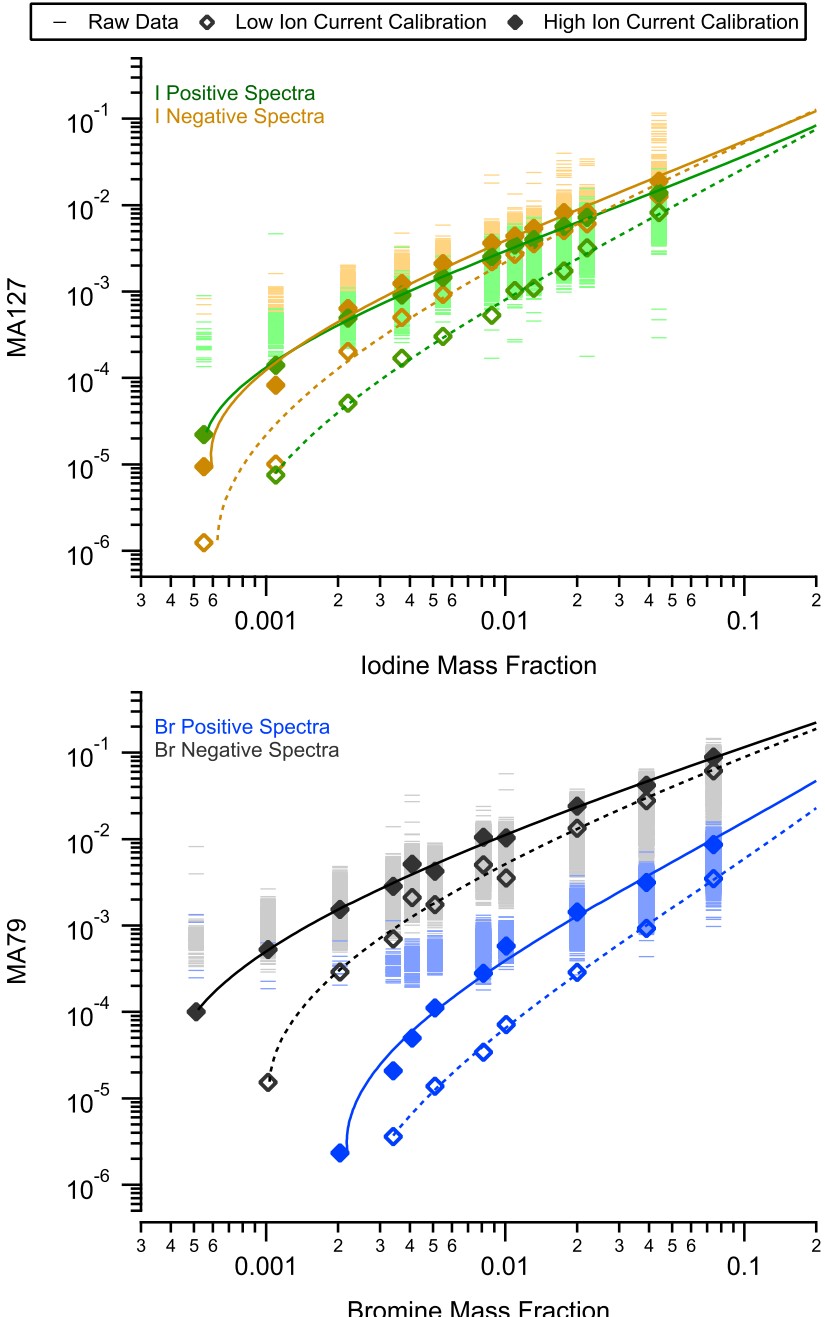

**Figure A2.** MA79 (MA127) as a function of bromine (iodine) mass fraction in a proxy for nSSA. Raw data are shown as horizontal bars, average signal per mass fraction as markers, and calibration curves as lines. Zero values are not shown because of the log scale, but are included when calculating the average values.



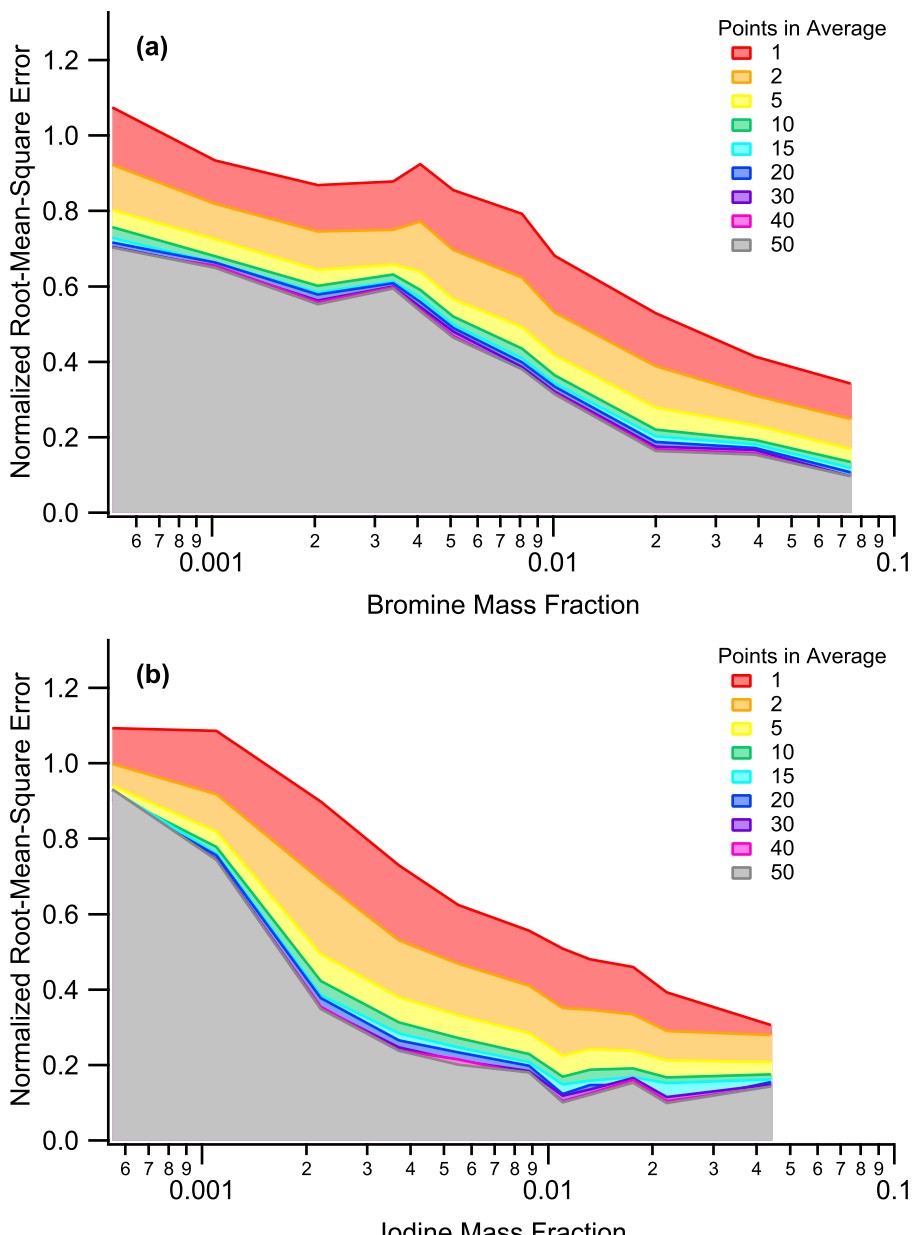

**Figure A3.** Normalized root-mean-square error as a function of (a) bromine mass fraction, and (b) iodine mass fraction. The number of particles used in an average are shown as overlaid graphs. Normalized room-mean-square errors are reduced with higher halogen mass fractions and with more particles used in an average.



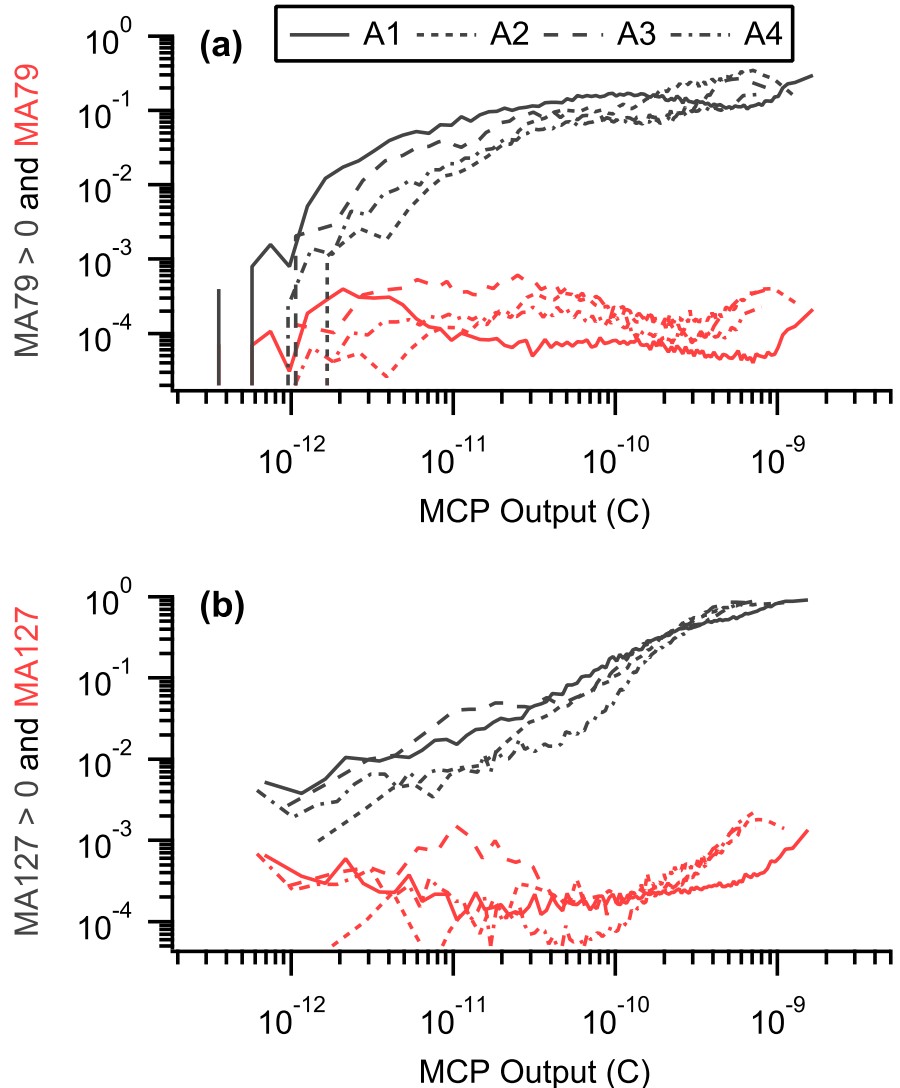

**Figure A4.** Number fraction of halogen-containing particles (black) and halogen peak area (red) vs. MCP output for (a) bromine (MA79) and (b) iodine (MA127). Number fractions are calculated as MA79 > 0 for bromine and MA127 > 0 for iodine. In this example, only positive spectra were used. Each line corresponds to a different ATom, *e.g.*, A1 = ATom-1



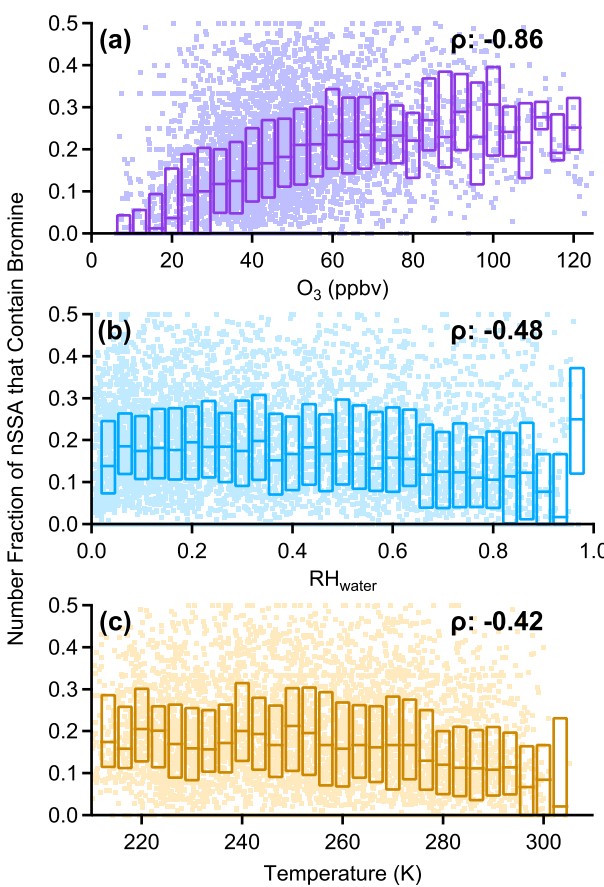

**Figure A5.** Box plots of the number fraction of nSSA that contain bromine (MA79 > 0) vs. (a) tropospheric $O_3$, (b) $RH_{water}$, and (c) temperature. Tropospheric air was defined as (altitudes > 8 km) $\wedge$ ($O_3$ > 100 ppbv) = 0. Raw data are dots, median values horizontal dashes, and the interquartile range is bound by the boxes. Median value Spearman's rho ($\rho$) values are reported in each panel.





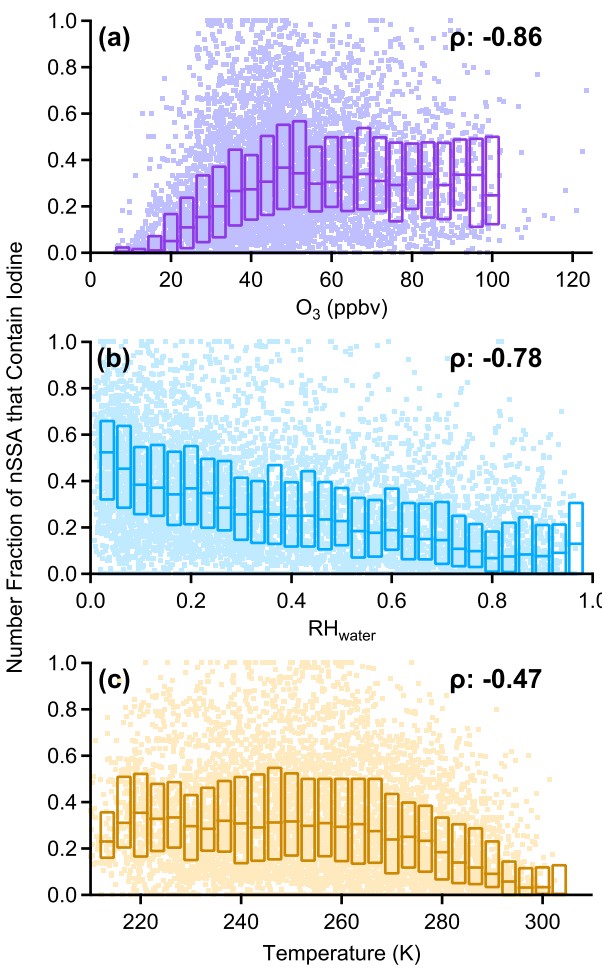

**Figure A6.** Box plots of the number fraction of nSSA that contain iodine (MA127 > 0) vs. (a) tropospheric $O_3$, (b) $RH_{water}$, and (c) temperature. Tropospheric air was defined as (altitudes > 8 km) $\wedge$ ($O_3$ > 100 ppbv) = 0. Raw data are dots, median values horizontal dashes, and the interquartile range is bound by the boxes. Median value Spearman's rho ($\rho$) values are reported in each panel.



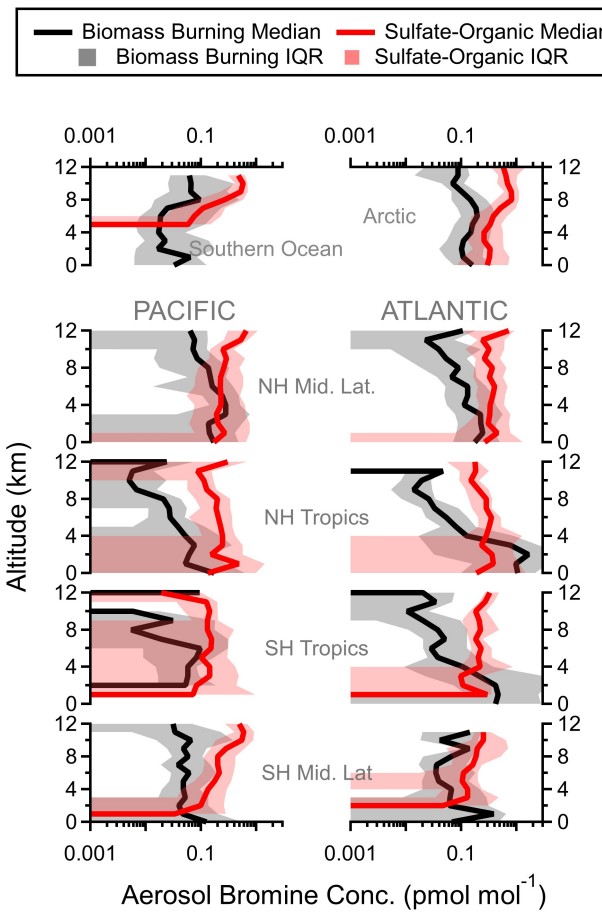

**Figure A7.** Regional vertical profiles of bromine mass (pmol mol$^{-1}$) in biomass burning and sulfate-organic-nitrate particles. Median values (solid lines) and interquartile ranges (shading) are from all four AToms. Latitude cuts for each region are in Table S1 in Schill et al. (2020).





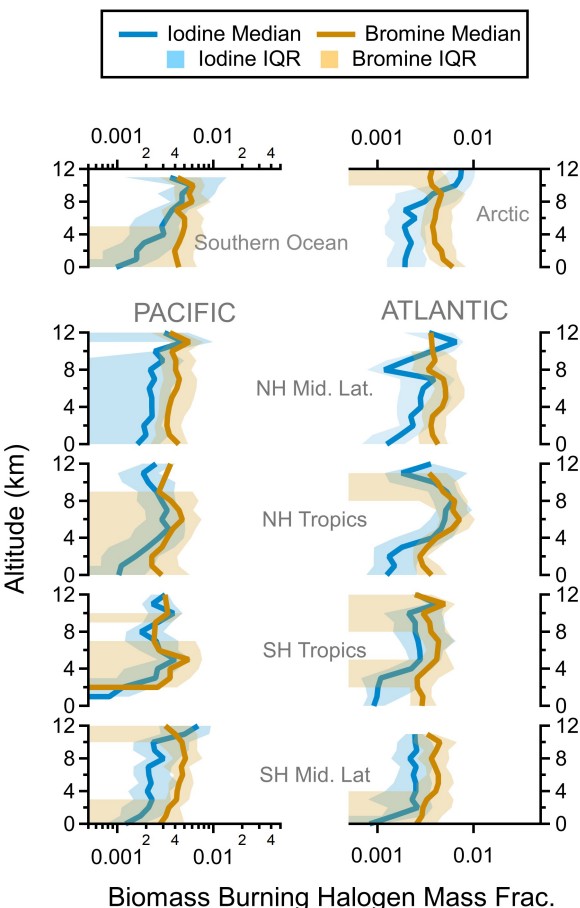

**Figure A8.** Regional vertical profiles of iodine and bromine mass fraction in biomass burning particles. Median values (solid lines) and interquartile ranges (shading) are from all four AToms. Latitude cuts for each region are in Table S1 in Schill et al. (2020).



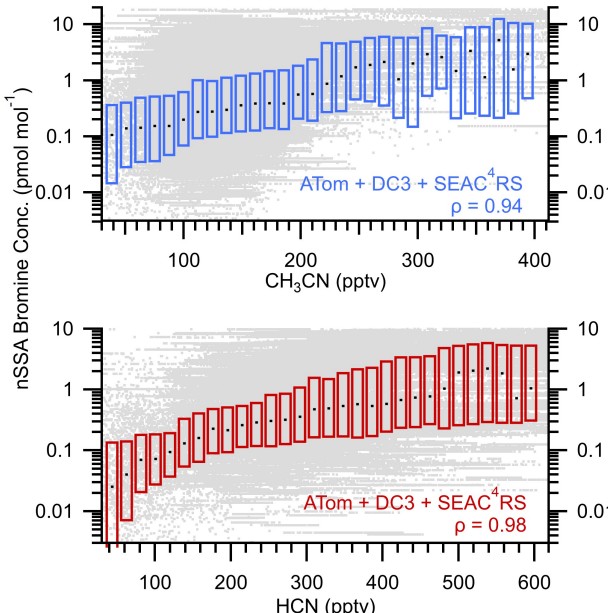

**Figure A9.** Box plots of nSSA bromine mass (pmol mol$^{-1}$) vs. gas-phase biomass burning tracers CH$_3$CN (pptv) and HCN (pptv). Raw data are gray dots, median values are black horizontal dashes, and the interquartile range is bound by the colored boxes. Median value Spearman's rho ($\rho$) are >0.94, which indicates near-monotonic relationship and suggests that biomass burning is a primary source of nSSA bromine mass.





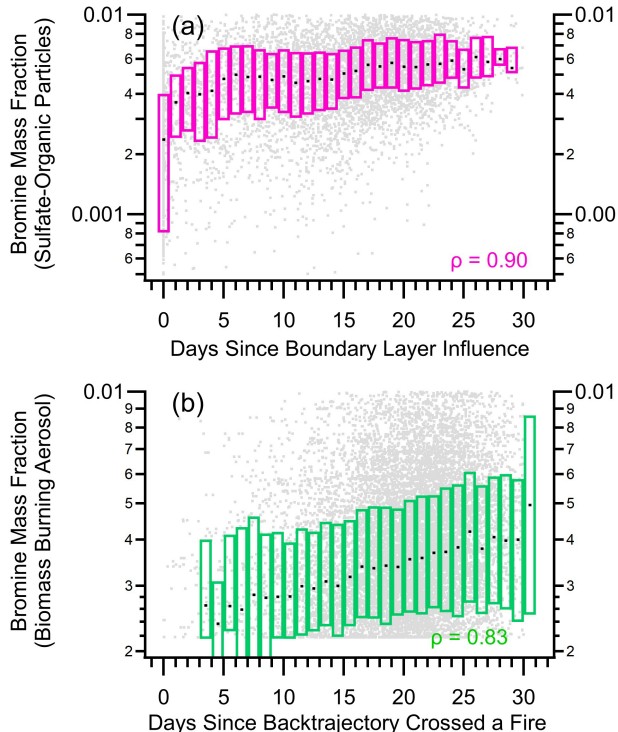

**Figure A10.** Box plots of nSSA bromine mass fractions vs. two back trajectory products, (a) the average number of days since a back trajectory was in the boundary layer, and (b) the number of days since a back trajectory crossed a fire. Raw data are gray dots, median values are black horizontal dashes, and the interquartile range is bound by the colored boxes. Median value Spearman's rho ($\rho$) are >0.83, which indicates near-monotonic relationship and suggests that biomass burning is a primary source of nSSA bromine mass.



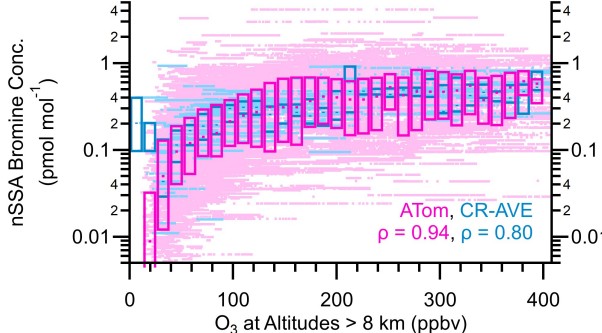

**Figure A11.** Box plots of nSSA bromine mass (pmol mol$^{-1}$) vs. O$_3$ for ATom (yellow) and CR-AVE (magenta). The median values of Spearman's rho ($\rho$) for ATom and CR-AVE are 0.94 and 0.80, respectively, which indicates a near-monotonic relationship and suggests that the stratosphere may be a source of nSSA bromine mass globally.



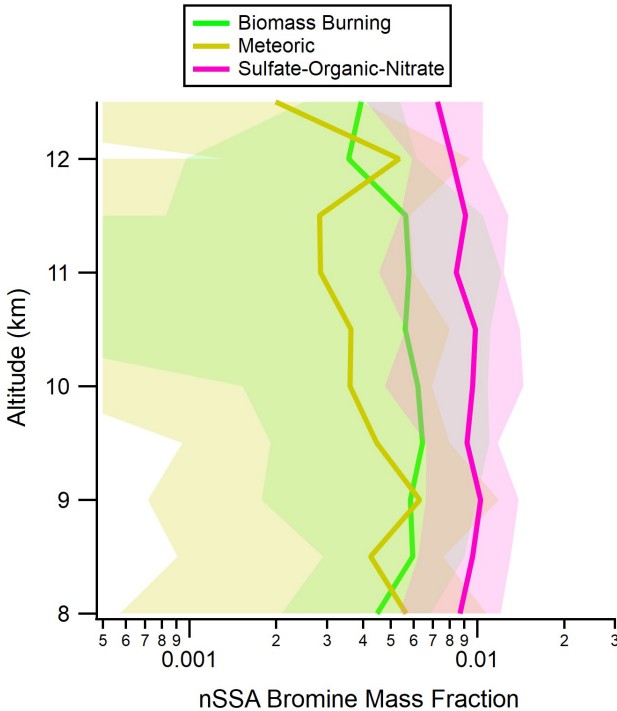

**Figure A12.** Vertical profiles of nSSA bromine mass fractions in the most common particle types found in the lower stratosphere during ATom: tropospherically sourced biomass burning and sulfate-organic-nitrate particles and stratospherically sourced meteoric-sulfuric-acid particles. Average values (solid lines) and standard deviations (shading) are from all four AToms. In this work. the lower stratosphere is defined as air masses above 8 km where the $[O_3]/[CO]$ ratio is greater than 3.



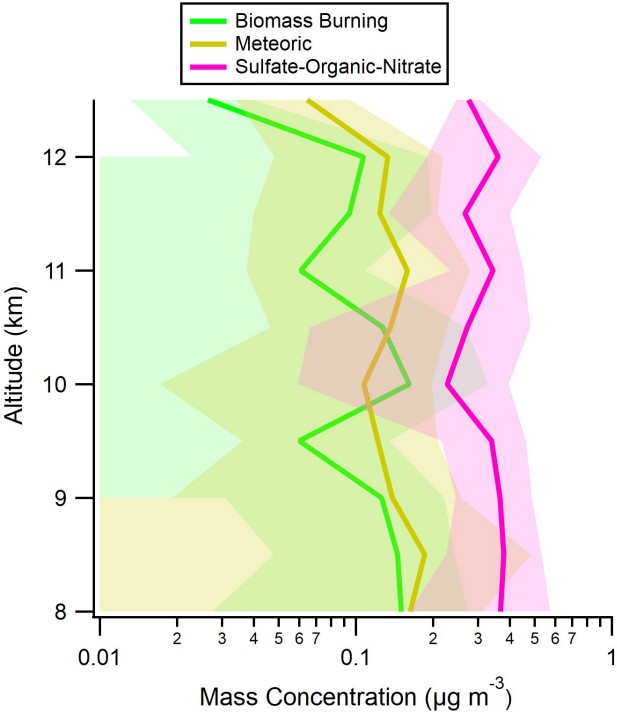

**Figure A13.** Vertical profiles of aerosol mass concentrations of biomass burning, sulfate-organic-nitrate, and meteoric-sulfuric-acid aerosol in the lower stratosphere during ATom. Average values (solid lines) and standard deviations (shading) are from all four AToms In this work, the lower stratosphere is defined as air masses above 8 km where the $[O_3]/[CO]$ ratio is greater than 3.





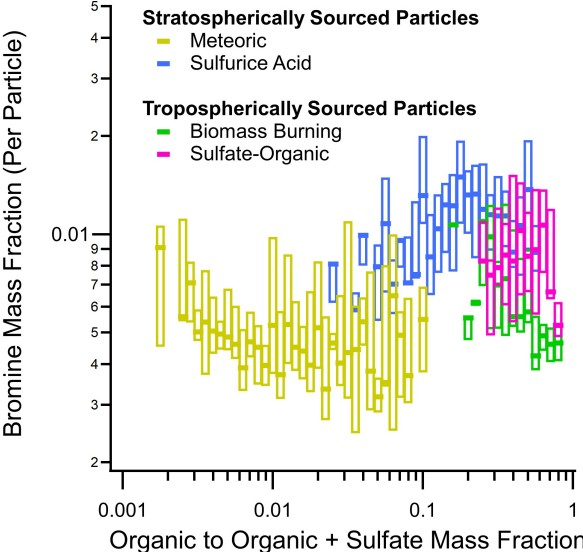

**Figure A14.** Box plots of bromine mass fraction in different particle types found in the lower stratosphere as function of organic-to-organic-plus-sulfate mass ratio from all four AToms. Of the four dominant particle types, two are tropospherically sourced (biomass burning and sulfate-organic-nitrate) and two are stratospherically sourced (meteoric-sulfuric-acid and sulfuric-acid). Horizontal dashes are median values and the boxes are interquartile ranges. In this work, the lower stratosphere is defined as air masses above 8 km where the $[O_3]/[CO]$ ratio is greater than 3.



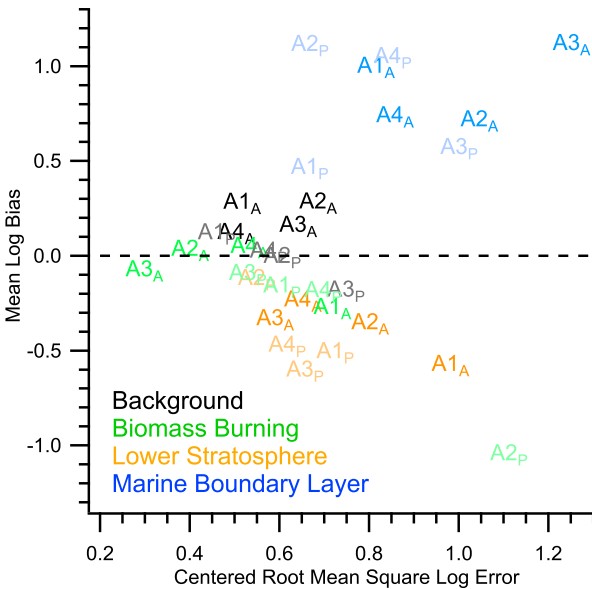

**Figure A15.** Mean log bias vs. centered root mean square log error (CRMSLE) of modelled nSSA iodine from GEOS-Chem 12.9.1. Both mean log bias and CRMSLE are calculated for each ATom and ocean basin (*e.g.*, $A1_P$ is ATom-1, Pacific Ocean and $A1_A$ is ATom-1, Atlanic Ocean). While the model is minimally biased in the background troposphere, the model tends to underestimated nSSA iodine in air influenced by biomass burning aerosol and in the lower stratosphere. Conversely, the model tends to overestimate the nSSA iodine in the marine boundary layer. Large biases are often associated with large CRMSLEs, which means that biases are not consistently offset, and could be due to missing processes instead of systematic errors.



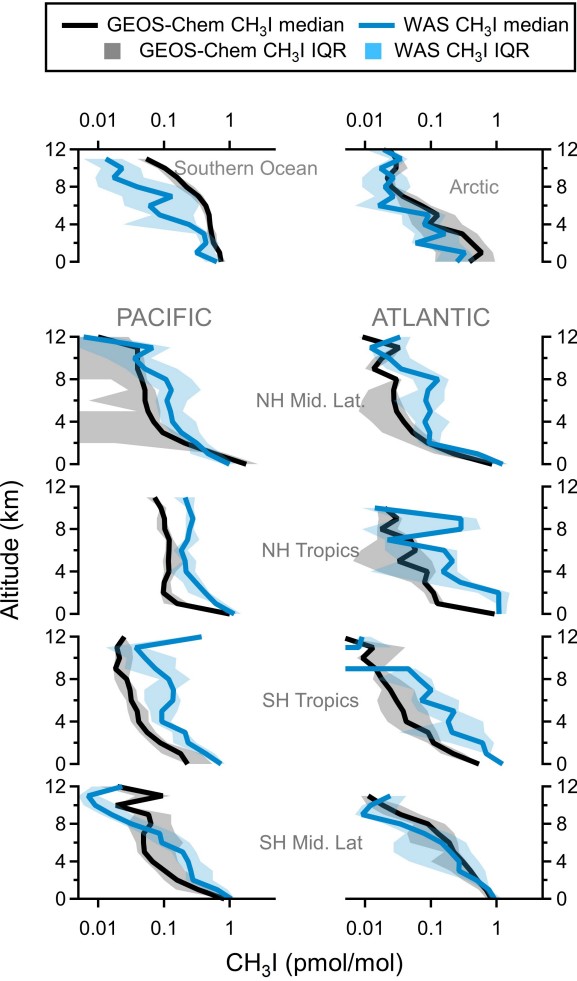

**Figure A16.** Regional vertical profiles of CH$_3$I (pmol mol$^{-1}$). Median values (solid lines) and interquartile ranges (shading) are from all four AToms. Latitude cuts for each region are in Table S1 in Schill et al. (2020).



*Author contributions.*  GPS, KDF, and DMM performed the PALMS measurements; CJW and CAB performed the size distribution measure-
ments; TS and MJE performed the GEOS-Chem modelling; EAR performed the back trajectory analysis; ECA, RSS, and AJH performed the
TOGA measurements; JP, TBR, CRT, and IB performed the $NO_yO_3$ measurements; DRB performed the WAS measurements, JPD and GSD
performed the DLH measurements. All measurement performers analyzed their data; GPS wrote the manuscript draft; All authors reviewed
and edited the manuscript

*Competing interests.*  The authors declare that they have no conflict of interest.

*Acknowledgements.*  The mission as a whole was supported by NASA's Earth System Science Pathfinder Program EVS-2 funding. Partici-
pation in ATom Mission flights by G.P.S., K.D.F., C.W., C.A.B. and D.M.M. was supported by NOAA climate funding (no. NNH15AB12I).
GPS, KDF, CJW, JP, TBR, CRT, and IB were supported by the NOAA Cooperative Agreement NA17OAR4320101. This material is based
upon work supported by the NSF National Center for Atmospheric Research, which is a major facility sponsored by the U.S. National
Science Foundation under Cooperative Agreement No. 1852977.



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
