# Peer review of "Widespread Trace Bromine and Iodine in Remote Tropospheric Non-Sea-Salt Aerosols"

_EGUsphere, 2024_

## Referee Comment (RC2)

Ozone catalytic destruction by halogen species and new particle formation and cloud condensation nuclei formation from iodine species are two important processes with potential climate implications. Most previous field studies focus on group-level measurements, while the vertical distributions of reactive halogens are much sparser. Many global chemical transport models have illustrated the potential role of bromine and iodine species in atmospheric chemistry and suggested that their impacts are ubiquitous. However, aircraft measurements have often been lacking to support these theses. The non-sea-salt aerosols (nSSA) bromine and iodine data presented in this paper, as part of the ATom campaign, are unique and severely needed for global model evaluations of the atmospheric chemistry impact of bromide and iodine species. Furthermore, several recent studies have highlighted the impact of iodine oxoacids in atmospheric new particle formation processes, which potentially contribute to cloud condensation nuclei formation and influence the aerosol indirect effect by affecting cloud properties. The data provided in this study will certainly be used by global models to constrain aerosol formation from iodine species.

This paper presents a comprehensive study showing the ubiquitous presence of nSSA containing trace amounts of bromine and iodine. One strength of the PALMS used in this study is that it separates aerosol types thereby can easily and directly associates bromine and iodine mass concentrations to different aerosol source types. The study finds that bromine and iodine are present in a significant fraction of nSSA, with biomass burning identified as a primary source. Additionally, it suggests that pervasive secondary sources of bromine and iodine are needed to explain their concentrations in nSSA. The finding of the correlation between biomass burning and elevated nSSA iodine and bromine is novel and represents a missing source of bromine and iodine in global models, as suggested by the comparison to GEOS-Chem results.

Overall, the paper is well-written, and the data presentation is clear and clean with no cluttered plots. The choice to place many similar plots in the appendix is encouraged to prevent overcrowding the main text. Therefore, I recommend this paper for publication in ACP after minor revisions.

Major:

One major problem with this paper is that the narrative emphasises two aspects: 1) bromine and iodine are ubiquitous in the nSSA throughout the atmosphere, and 2) bromine and iodine mass concentrations are negligible compared to organics and sulfate. I believe the discussion should not stop here. A critical difference between particulate iodine/bromine and sulfate/organics is that iodine and bromine are actively involved in heterogeneous processes, thereby are efficiently recycled, whereas sulfate and organics primarily remain in the particles once condensed. In other words, mass concentration is not a sufficient metric for characterising the importance of halogen species. It would be a pity for this study to stop at this point. The study integrates the GEOS-Chem simulation, and it should be relatively straightforward for the authors to delve further, such as by correlating iodine mass concentrations with its impact (or

the potential impact, by any reasonable metric) on odd oxygen. This would help readers understand the significance of low mass concentrations in practical terms. Is it important or not? As the authors pointed out, many factors influence the impact of iodine on odd oxygen, which has been previously explored. However, providing such a calculation, correlating mass concentration to atmospheric impact along the tracks of ATom, is critical and has the potential to be widely cited (I would for sure cite this in upcoming studies).

It would be great if the authors could present the I+IO data in the appendix. The CH3I measurement from the GEOS-Chem simulation captures well what ATom provides. Including I+IO as a metric for future evaluation of iodine chemistry and aerosol formation would be highly valuable.

After obtaining these data in the authors' preferred formats, please correlate the bromine/iodine mass concentrations with their atmospheric implications throughout the manuscript, especially in the abstract and conclusions. This will undoubtedly enhance the manuscript's significance and "citability."

Another major problem is the insufficient emphasis on the importance of heterogeneous processes, i.e., the recycling of halogen species between the gas and particulate phases. As detailed below, several raised hypotheses could at least be partially explained by these heterogeneous processes. While determining the exact reasons for the under/over-estimated bromine/iodine levels of GEOS/Chem, and gas-to-particle partitioning processes in the measurements, are outside the scope of this study, mentioning this possibility would provide a more complete discussion.

Minor:

Suggested title: Widespread Trace Bromine and Iodine in Remote Tropospheric Non-Sea-Salt Aerosols. I believe this will highlight the key message of this study. However, I will leave this to the authors to decide whether to adopt the suggestion or not.

Lines 6-7: it should also note that low concentrations of halogens can already introduce significant impact.

Line 20: Sherwen et al. 2016b should be quoted first as Sherwen et al. 2016a.

Lines 42-44: the lines are a bit confusing. The iodine contribution to secondary aerosols is direct but the bromide contribution is indirect through its a bit limited capability (compared with chlorine radicals) to react with organic species. It should be clarified.

Lines 45-47: it should be noted that the prevailing evidence currently shows iodine oxoacids play key roles in iodine secondary aerosol formation processes, while iodine oxides play relatively smaller roles. However, since I believe the reviewer 1 might have a differing opinion, it is the best for this study to give more generic expression such as "iodine species" to pass the review process.

Lines 49-51: Finkenzeller et al., 2023 does not show iodic acid can nucleate on its own. Please rephrase the reference to their results.

Line 58: please convert ng m−3 to pmol mol −1 which is frequently used in this study. Also line 58: please provide the measured range of the Koenig et al. (2020).

Lines 133-134: I do not understand how masses 95 and 97 can affect the bromide measurement? Is this instrument specific problem? Please specify.

From the section 2.2, it seems the systematic errors of the reported bromine and iodine concentrations are more likely the lower limit than higher limit. It would be great if such systematic error can be reflected in the abstract and conclusions where such numbers are quoted.

Lines: 254-255: This is interesting. However, the explanation is not convincing enough since gaseous and particulate bromine and iodine are constantly exchanging through heterogenous processes. Therefore, this cannot be a single way of accumulation of mass. It is also likely that the dry air prohibits such exchange, therefore resulting in net accumulation of bromide and iodine in aerosols.

Lines 330-336: Is there a correlation between days since boundary layer influence and relative humidity? Could the authors confirm this? Additionally, could the authors discuss whether such findings are due to bromine/iodine accumulation or shifts in gas/particle partitioning caused by changes in temperature and relative humidity?

Lines 337-344: following the same as the last comment, could the change in RH also a possibility?

Lines 351-353: going into the same line - difficult for me to imagine how RH is going to prevent condensable iodine species to condense onto particles, as laboratory experiments have pointed to a kinetic condensation of species such as $HIO_3$, which is the dominant mass contributor in at least the MBL[1]. However, the reverse may be more true: high RH may promote the release of iodine from particles, thereby reducing the iodine particle mass concentration.

Lines365-366: another line of evidence supports potential RH effect.

Lines 367-383: meteoric aerosols presumably contain a lot of iron which promotes the conversion of iodide to molecular iodine[2]. This may be another reason for the strikingly lower aerosol iodine in meteoric aerosols than the sulfate-organic-nitrate / biomass burning aerosols.

Lines 384-393: along the same line, is it possible to check the correlation between metals such as iron vs. iodine content?

Lines 412-413: inorganic emissions of what? Presumably halogens?

Lines 459-460: The statement that water reduces iodine particle formation in the MBL is unfortunately incorrect. Experiments conducted under atmospheric conditions generally indicate a positive enhancement of particle nucleation rates with increasing water content. Specifically for iodine aerosol nucleation, recent evidence suggests that relative humidity above 2% has negligible impact on iodine aerosol nucleation in the marine boundary layer[3]. Please remove this statement. Furthermore, the high levels of iodine in nSSA are also likely attributable to missing heterogeneous chemistry processes.

Lines 469-472: could the authors please specify what are the major iodine components in nSSA in GEOS-Chem for a comparison with the ATom measumements? Ideally it should be quantitative.

Lines 469-477 are difficult to follow, and more context is needed for general readers regarding what is consistent and what is not consistent between the conclusions of this study and Koenig et al. 2020[4]. While reformulating the discussion, it is worth mentioning that the higher iodate

in the LS would already indicates lower regeneration of iodine to the gas phase, thereby may lead to lower gas phase reactive iodine, assuming similar total nSSA iodine in the LS and UT.

**References:**

(1) He, X.-C.; Tham, Y. J.; Dada, L.; Wang, M.; Finkenzeller, H.; Stolzenburg, D.; Iyer, S.; Simon, M.; Kürten, A.; Shen, J.; Rörup, B.; Rissanen, M.; Schobesberger, S.; Baalbaki, R.; Wang, D. S.; Koenig, T. K.; Jokinen, T.; Sarnela, N.; Beck, L. J.; Almeida, J.; Amanatidis, S.; Amorim, A.; Ataei, F.; Baccarini, A.; Bertozzi, B.; Bianchi, F.; Brilke, S.; Caudillo, L.; Chen, D.; Chiu, R.; Chu, B.; Dias, A.; Ding, A.; Dommen, J.; Duplissy, J.; El Haddad, I.; Gonzalez Carracedo, L.; Granzin, M.; Hansel, A.; Heinritzi, M.; Hofbauer, V.; Junninen, H.; Kangasluoma, J.; Kemppainen, D.; Kim, C.; Kong, W.; Krechmer, J. E.; Kvashin, A.; Laitinen, T.; Lamkaddam, H.; Lee, C. P.; Lehtipalo, K.; Leiminger, M.; Li, Z.; Makhmutov, V.; Manninen, H. E.; Marie, G.; Marten, R.; Mathot, S.; Mauldin, R. L.; Mentler, B.; Möhler, O.; Müller, T.; Nie, W.; Onnela, A.; Petäjä, T.; Pfeifer, J.; Philippov, M.; Ranjithkumar, A.; Saiz-Lopez, A.; Salma, I.; Scholz, W.; Schuchmann, S.; Schulze, B.; Steiner, G.; Stozhkov, Y.; Tauber, C.; Tomé, A.; Thakur, R. C.; Väisänen, O.; Vazquez-Pufleau, M.; Wagner, A. C.; Wang, Y.; Weber, S. K.; Winkler, P. M.; Wu, Y.; Xiao, M.; Yan, C.; Ye, Q.; Ylisirniö, A.; Zauner-Wieczorek, M.; Zha, Q.; Zhou, P.; Flagan, R. C.; Curtius, J.; Baltensperger, U.; Kulmala, M.; Kerminen, V.-M.; Kurtén, T.; Donahue, N. M.; Volkamer, R.; Kirkby, J.; Worsnop, D. R.; Sipilä, M. Role of Iodine Oxoacids in Atmospheric Aerosol Nucleation. *Science* **2021**, *371* (6529), 589–595. https://doi.org/10.1126/science.abe0298.

(2) Fudge, A. J.; Sykes, K. W. 25. The Reaction between Ferric and Iodide Ions. Part I. Kinetics and Mechanism. *J. Chem. Soc. Resumed* **1952**, 119. https://doi.org/10.1039/jr9520000119.

(3) Rörup, B.; He, X.-C.; Shen, J.; Baalbaki, R.; Dada, L.; Sipilä, M.; Kirkby, J.; Kulmala, M.; Amorim, A.; Baccarini, A.; Bell, D. M.; Caudillo-Plath, L.; Duplissy, J.; Finkenzeller, H.; Kürten, A.; Lamkaddam, H.; Lee, C. P.; Makhmutov, V.; Manninen, H. E.; Marie, G.; Marten, R.; Mentler, B.; Onnela, A.; Philippov, M.; Scholz, C. W.; Simon, M.; Stolzenburg, D.; Tham, Y. J.; Tomé, A.; Wagner, A. C.; Wang, M.; Wang, D.; Wang, Y.; Weber, S. K.; Zauner-Wieczorek, M.; Baltensperger, U.; Curtius, J.; Donahue, N. M.; El Haddad, I.; Flagan, R. C.; Hansel, A.; Möhler, O.; Petäjä, T.; Volkamer, R.; Worsnop, D.; Lehtipalo, K. Temperature, Humidity, and Ionisation Effect of Iodine Oxoacid Nucleation. *Environ. Sci. Atmospheres* **2024**, 10.1039.D4EA00013G. https://doi.org/10.1039/D4EA00013G.

(4) Koenig, T. K.; Baidar, S.; Campuzano-Jost, P.; Cuevas, C. A.; Dix, B.; Fernandez, R. P.; Guo, H.; Hall, S. R.; Kinnison, D.; Nault, B. A.; Ullmann, K.; Jimenez, J. L.; Saiz-Lopez, A.; Volkamer, R. Quantitative Detection of Iodine in the Stratosphere. *Proc. Natl. Acad. Sci.* **2020**, *117* (4), 1860–1866. https://doi.org/10.1073/pnas.1916828117.

---

## Author Comment (AC1)

**Response to "Reviewer Comments"**

Gregory P. Schill

7 September 2024

**1 Anonymous Referee 1**

Schill et al. present global-scale aircraft measurements of bromine and iodine in accumulation mode non-sea-salt aerosol carried out using the PALMS instrument. It is worth highlighting that measurements of iodine, and specially of bromine in aerosol are scarce and almost exclusively confined to near-surface campaigns. Therefore, the dataset presented in this paper is extremely valuable. The measurements show that iodine and bromine are common in nSSA, although their mass concentrations are low. Analysis of the data indicates two sources of iodine and bromine in the troposphere: a primary source related to biomass burning and a secondary source linked to uptake of gas phase compounds, where ozone is likely to play a role in converting reactive halogens into their aerosol-bound forms, and the halogen mass fraction increases as aerosols age. In the stratosphere, the iodine and bromine mass concentrations are also observed to increase with increasing ozone, but the highest concentrations are found in organic-rich aerosols of tropospheric origin, which leads the authors to conclude that organics play a role in retaining iodine and bromine in nSSA (or are a proxy of some process helping retention). This extensive dataset including measurements from the MBL to the UT-LS for different latitudes and longitudes provides a benchmark against which the performance of global models including halogen aerosol chemistry can be tested. The manuscript presents a comparison of the iodine observations with simulations from the GEOS-Chem model, even though some processes such as the primary biomass source of iodine are not implemented in the model yet. The simulations compare reasonably with the measurements in the background free troposphere, indicating that the secondary source is essentially captured. Elsewhere (MBL, tropospheric biomass burning plumes, stratosphere) the lack of agreement is expected because the relevant processes are not included in the model.

The instrument characterization is sound and the measurements are of high quality. The paper is well written and the conclusions are supported by the data. I recommend publication after minor revisions. I have compiled

> below a list of comments that the authors may want to consider in order to improve their manuscript.

Thank you for your comment. We are delighted to hear that you think the paper and measurements are of high quality.

> Page 2, line 31: Cuevas et al (2018) reported both things first. Please cite Cuevas et al. 2018 also for the enhanced ozone inorganic iodine source alongside Legrand et al. 2018.

Done. We have added the Cuevas et al., 2018 reference alongside the Legrand et al., 2018 reference for the enhanced ozone inorganic iodine source. Thank you for bringing this to our attention.

> Page 2, line 41: a few years after Dean et al. (1963), Duce et al. (JGR, 88, 1983) provided evidence showing that marine aerosols are enriched with iodine largely because they sorb gas-phase iodine

Thank you for providing us with this reference. We have changed the text to read "... or from fast gas- or mutli-phase chemistry (Duce et al., 1983), or both."

> Page 2, lines 42-43: it would be helpful for the non-expert to provide a brief definition of nSSA, mentioning the relevant nSSA particle size range compared to SSA. This would help to understand better the statements in lines 52-53 in page 3.

Thank you for your comment. To help clarify this discussion for non-experts, we have added the following text to Line 37:

"... (SSA), which are produced from wave breaking or bubble bursting at the ocean surface,"

We have also added the following to Line 52:

"SSA; while nSSA differs from SSA in that they are typically non-refractory and smaller in diameter (fine mode), it is difficult to cleanly separate nSSA from SSA based on size alone. Nonetheless, filter-based ..."

Lastly we have added the following text to Line 54:

"Single-particle techniques can separate nSSA from SSA based on their chemical composition. Qualitative measurements from the single-particle mass spectrometer Particle Analysis by Laser Mass Spectrometry ..."

Page 3, line 63. For the non-expert, it would be helpful to explain briefly why the study focuses on nSSA and how is nSSA discriminated. The discrimination of nSSA particles is explained in page 5, lines 121-123 but I think this should be explained earlier in the text.

Please see our response to the reviewer's comment about Lines 42-43.

Page 4, line 111. "An inherent assumption of this technique is that the particle type fractions are constant within each bin." Is this a reasonable assumption?

This assumption has been tested in detail by Froyd et al., (2019). We have added the following text to that paragraph: "For ATom, we have found this assumption to be reasonable for the 4 bins described above (Froyd et al., 2019)."

Page 4 line 110 and elsewhere: mass concentrations are given in ug m-3 here, but in the abstract are given in pmol mol-1. Then again in page 10, line 258, the median and IQR of mass concentration for iodine and bromine are given in ng m-3, and are then converted into pmol mol-1. From this point on, concentrations are given in pmol mol-1. However, this is a mixing ratio "unit" (actually dimensionless), not a mass concentration unit (mass per volume of air). This can be confusing, so I would suggest that the authors define precisely what they mean by mass concentration and how they convert it to mixing ratio, and then to keep consistence in using either concentrations or mixing ratios throughout the text. I would suggest not using concentration and mixing ratio interchangeably and modifying accordingly the axes tittles in a number of figures indicating concentrations in pmol mol-1.

The reviewer makes a good point here; however, to accommodate a wide readership, we have decided to keep to the popular conventions of reporting aerosol halogens in pmol mol$^{-1}$ and aerosol concentration in $\mu g\ m^{-3}$. This allows for the most direct comparison of our measurements to the literature. Additionally, using pmol mol$^{-1}$ allows readers to compare aerosol halogen concentrations with the gas-phase bromine and iodine budgets. We do, however, agree with the reviewer that it can be confusing to switch between the concentration units; thus we have added the following sentences to Line 110:

"We convert between pmol mol$^{-1}$ and ng sm$^{-3}$ by assuming that the atmosphere is an ideal gas and that the MW of bromine and iodine are 79.9 g mol$^{-1}$ and 126.9 g mol$^{-1}$, respectively."

Furthermore, we want to be clear that all of our mass concentrations are reported at standard conditions. Thus, in all instances in the paper, we have changed m$^{-3}$ to sm$^{-3}$, and defined sm$^{-3}$ upon its first use. Our other concern is to

ensure that the reader knows we're talking about volume mixing ratios instead of mass mixing ratios, which is why we use pmol mol$^{-1}$ instead of ppt.

We also concede that it is incorrect to call the mixing ratio a mass concentration. All instances of this in the paper have been removed; however, we will continue to call mixing ratios "concentrations," as that is also commonly done in the atmospheric trace-gas literature.

> Page 6, lines 151-152: in what physical magnitude is given the MCP output? What unit is C?

We note that we could be more clear about the MCP output. We have modified to text to read "... high total ion signal [high MCP output, units Coulomb (C)] ..."

> Page 8, section 3.1: A comparison with previous aerosol bromine and iodine measurements is absent from the discussion. For example, do the MBL ATom measurements of aerosol iodine show similar geographical trends to near-surface TI measurements?

We thank the reviewer for this excellent point. We have added the following text to Section 3.1: "These IQRs are lower than the range of total iodine concentrations that Gómez Martín et al. (2021) calculated for the remote MBL cruises TransBrom and AMT21 (0.57 - 1.05 pmol mol$^{-1}$), but it should be noted that these cruise measurements are not able to separate nSSA from SSA. This bias will be exacerbated for total bromine, since the natural seawater concentration of bromine is orders of magnitude higher than iodine, and much more bromine will be found naturally in sea salt aerosol. Thus, we will not attempt to make a comparison of bulk measurements to our bromine measurements here."

> Page 9, lines 246-249 and Figs A5 and A6: using $O_3$ mixing ratio in ppbv rather than absolute concentration may blur the link between halogen-containing particle fraction and ozone as a limiting reactant. Within the troposphere the pressure changes significantly, which means that the same mixing ratio at different altitudes corresponds to different concentrations. By contrast, Figs. 7 and 9 are fine because in those halogen mixing ratios are plotted vs trace gas mixing ratio, which effectively is the same as plotting concentration vs concentration.

The reviewer makes an excellent point here. We have changed this sentence to read: "Thus, low $O_3$ may be an important condition for time periods when the conversion of reactive bromine and iodine to their aerosol-bound forms is inhibited, or that aerosol-bound bromine and iodine may titrate $O_3$ through heterogeneous reactions that liberate these halogens from the aerosol phase; however, it is difficult to draw a quantitative conclusion from these correlations

because number fraction and pmol mol$^{-1}$ are inherently different units that scale differently with pressure."

> Page 9, line 248: would it be possible as well that from a certain $O_3$ concentration, heterogeneous $O_3$ reactions on the aerosol surface like iodide + $O_3$ recycle halogens back to the gas phase?

Please see out response to your previous comment.

> Page 15, figure 7: is pptv here not the same as pmol mol-1?

It is, we have changed the axes and titles to now read: pmol mol$^{-1}$ (= pptv). We keep the reference to pptv as the trace-gas community typically uses it. The same has been done for Figure A9. We have also added (= pptv) to our first use of pmol mol$^{-1}$ for readers who are more used to this notation.

> Page 18, Figure 9: : is ppbv here not the same as nmol mol-1?

It is, we have changed the axes and titles to now read: nmol mol$^{-1}$ (= ppbv). We keep the reference to ppbv as the trace-gas community typically uses it. The same has been done for Figure A11.

> Page 18, line 380: "It has also been shown from aerosols collected at the Mace Head research station that over 90% of soluble aerosol iodine is organically bound, with the rest being iodide or iodate (Gilfedder et al., 2008)". This has been observed only at Mace Head and in a single campaign (MAP 2006), so it should be considered as a special case. It has been noted that long ultrasonication times employed in the iodine extraction process in that work may have influenced speciation. But it is remarkable that the high SOI fractions reported for MAP 2006 are concurrent with extremely high values of total soluble iodine. The ground-based campaign and the associated cruise reported the highest median values of the total iodine aerosol record (44 ng m-3) (Gómez Martín et al., 2021). So even though this is possibly a special case, the two observations appear to support the authors' argument that SOI helps retaining iodine in aerosol. Elsewhere on the surface, soluble organic iodine makes up between 40 and 60% of fine aerosol iodine (Gómez Martín et al. 2022).

We thank the reviewer for bringing this citation to our attention. We have changed that sentence to now read: "Furthermore, a comprehensive data set of field observations of iodine speciation in marine aerosol has shown that the soluble iodine content of fine aerosol (PM1) is approximately 50% soluble organic iodine by mass (Gómez Martín et al., 2022), with one study at the Mace Head head research station showing that ..."

> Page 20, line 416: I would not say that $HIO_3$ formation is a "known" process. The $HIO_3$ molecule has been observed in the atmosphere and appears to be an important and ubiquitous iodine carrier, but its formation is not understood. Finkenzeller et al. (2023) have proposed a theoretical mechanism involving I2O2, $O_3$ and H2O that looks unfeasible on thermochemical grounds. In any case, $HIO_3$ is likely to form somehow from IxOy + H2O, and either $HIO_3$ or IxOy, or both, are lost to aerosol, so it should not make a huge difference as long as an effective uptake process is included in the model.

We agree with the reviewer that our language may have been too strong here. We have rewritten this sentence to read "many known processes [e.g., heterogeneous / multi-phase reactions (Saiz-Lopez et al., 2012)] and some that have been recently proposed [e.g., iodine oxoacid new particle formation (He et al., 2021) and halogen solvation by organics (Solomon et al., 2023)];"

> Page 22, line 455-456: regarding the overestimation of nSSA iodine by a factor of 7, is it possible that the iodine recycling mechanism is incomplete or missing? For example, could gas phase ozone react heterogeneously with aerosol iodide to form HOI and I2?

The reviewer brings up a good point. We should point out that, in the model, HI and the higher iodine oxides are locked into the aerosol phase after uptake. We have added the following sentence to the Line 455: "Also, the model does not include any heterogeneous recycling reactions that would react with HI or higher iodine oxides once they are in the aerosol phase."

**2    Xu-Cheng He**

> Ozone catalytic destruction by halogen species and new particle formation and cloud condensation nuclei formation from iodine species are two important processes with potential climate implications. Most previous field studies focus on group-level measurements, while the vertical distributions of reactive halogens are much sparser. Many global chemical transport models have illustrated the potential role of bromine and iodine species in atmospheric chemistry and suggested that their impacts are ubiquitous. However, aircraft measurements have often been lacking to support these theses. The non-sea-salt aerosols (nSSA) bromine and iodine data presented in this paper, as part of the ATom campaign, are unique and severely needed for global model evaluations of the atmospheric chemistry impact of bromide and iodine species. Furthermore, several recent studies have highlighted the impact of iodine oxoacids in atmospheric new particle formation processes, which potentially contribute to cloud condensation nuclei formation and influence the aerosol indirect effect by affecting cloud properties.

The data provided in this study will certainly be used by global models to constrain aerosol formation from iodine species.

This paper presents a comprehensive study showing the ubiquitous presence of nSSA containing trace amounts of bromine and iodine. One strength of the PALMS used in this study is that it separates aerosol types thereby can easily and directly associates bromine and iodine mass concentrations to different aerosol source types. The study finds that bromine and iodine are present in a significant fraction of nSSA, with biomass burning identified as a primary source. Additionally, it suggests that pervasive secondary sources of bromine and iodine are needed to explain their concentrations in nSSA. The finding of the correlation between biomass burning and elevated nSSA iodine and bromine is novel and represents a missing source of bromine and iodine in global models, as suggested by the comparison to GEOS-Chem results.

Overall, the paper is well-written, and the data presentation is clear and clean with no cluttered plots. The choice to place many similar plots in the appendix is encouraged to prevent overcrowding the main text. Therefore, I recommend this paper for publication in ACP after minor revisions.

Thank you for your comments! We are delighted to hear that you think the paper is well written and the data presentation is clear and clean with no cluttered results.

One major problem with this paper is that the narrative emphasises two aspects: 1) bromine and iodine are ubiquitous in the nSSA throughout the atmosphere, and 2) bromine and iodine mass concentrations are negligible compared to organics and sulfate. I believe the discussion should not stop here. A critical difference between particulate iodine/bromine and sulfate/organics is that iodine and bromine are actively involved in heterogeneous processes, thereby are efficiently recycled, whereas sulfate and organics primarily remain in the particles once condensed. In other words, mass concentration is not a sufficient metric for characterising the importance of halogen species. It would be a pity for this study to stop at this point. The study integrates the GEOS-Chem simulation, and it should be relatively straightforward for the authors to delve further, such as by correlating iodine mass concentrations with its impact (or the potential impact, by any reasonable metric) on odd oxygen. This would help readers understand the significance of low mass concentrations in practical terms. Is it important or not? As the authors pointed out, many factors influence the impact of iodine on odd oxygen, which has been previously explored. However, providing such a calculation, correlating mass concentration to atmospheric impact along the tracks of ATom, is critical and has the potential to be widely cited (I would for sure cite this in upcoming studies).

After obtaining these data in the authors' preferred formats, please cor-

> relate the bromine/iodine mass concentrations with their atmospheric implications throughout the manuscript, especially in the abstract and conclusions. This will undoubtedly enhance the manuscript's significance and "citability."

The reviewer makes an excellent point here. We agree that giving some context to the aerosol halogen concentrations would increase the citability of the paper; however, we feel that quantifying the effect of aerosol halogens on something like the odd oxygen budget is outside of the scope of the paper. That being said, we have added the following sentence to the end of the paper:

"Thus, global chemistry-climate models that wish to explore the full extent of the impact of ubiquitous, trace bromine and iodine in nSSA on atmospheric constituents like odd oxygen, may first need to update their emission inventories and chemical schemes."

There are two main reasons that we feel that quantifying the effect of aerosol halogens on something like the odd oxygen budget is outside of the scope of this paper:

(1) To truly quantify this effect, we would have to do another full model run that turns off iodine aerosol partitioning and then do an on-minus-off calculation; however, we know that in the areas of the greatest interest for this paper (MBL, biomass burning plumes, stratosphere), the model does not replicate the results well.

(2) Even if we conducted more model runs to do an on-minus-off calculation, there is little guidance from the data on how we should treat the heterogeneous chemistry of our aerosol-bound iodine. Iodide is thought to be efficiently recycled back to the gas phase, and some mechanisms for iodate reduction have been proposed as well, but there are no clear mechanisms for organically bound iodine to react with odd oxygen heterogeneously. Additionally, in this paper, Figure 7 suggests that particles accumulate iodine as they age, suggesting that heterogeneous chemistry is slower than reactions that make aerosol-bound iodine. Thus, the odd oxygen results from an on-minus-off calculation would vary from a negative effect to a positive effect, and the sign of the effect would depend on how we defined / prescribed the heterogeneous chemistry reactions in the model.

Finally, in this paper, we believe that the citability of the paper derives from our discovery of three "sources" of aerosol halogens, namely a primary biomass burning source, a pervasive secondary source, and a stratospheric "source" or an enhancement of aerosol halogens in tropspherically sourced particles in the lower stratosphere. To accentuate this, we show that a state-of-the-art model replicates the pervasive secondary source, but it missing the biomass burning and lower stratospheric source. We also show that the model also does poorly in the MBL, which is interesting because that is where most of the measurements

of aerosol iodine to date have been taken. We have emphasized these points in our conclusion, but we believe that the sentence that we added at the end of paper emphasizes this point.

> Another major problem is the insufficient emphasis on the importance of heterogeneous processes, i.e., the recycling of halogen species between the gas and particulate phases. As detailed below, several raised hypotheses could at least be partially explained by these heterogeneous processes. While determining the exact reasons for the under/over-estimated bromine/iodine levels of GEOS/Chem, and gas-to-particle partitioning processes in the measurements, are outside the scope of this study, mentioning this possibility would provide a more complete discussion.

The reviewer makes another excellent point. We address this comment in our response to a string of minor comments below.

> Suggested title: Widespread Trace Bromine and Iodine in Remote Tropospheric Non-Sea-Salt Aerosols. I believe this will highlight the key message of this study. However, I will leave this to the authors to decide whether to adopt the suggestion or not.

Changed. Thank you for the suggestion!

> Lines 6-7: it should also note that low concentrations of halogens can already introduce significant impact.

The reviewer makes a good point. We have added the following qualifier to the abstract "... low but potentially important, 0.11-0.57 ..."

> Lines 42-44: the lines are a bit confusing. The iodine contribution to secondary aerosols is direct but the bromide contribution is indirect through its a bit limited capability (compared with chlorine radicals) to react with organic species. It should be clarified.

We thank the reviewer for pointing this out. We have added the following text for clarification: "... Badia 2019). Thus, the bromine aerosol contribution needs reactive organic species to contribute to secondary aerosol; iodine, however, can contribute to secondary aerosol through self reactions."

> Lines 45-47: it should be noted that the prevailing evidence currently shows iodine oxoacids play key roles in iodine secondary aerosol formation processes, while iodine oxides play relatively smaller roles. However, since I believe the reviewer 1 might have a differing opinion, it is the best for this study to give more generic expression such as "iodine species" to pass the review process.

Noted. We have added the phrase "or iodine oxoacids ($HIO_X$, X = 2,3)"

> Lines 49-51: Finkenzeller et al., 2023 does not show iodic acid can nucleate on its own. Please rephrase the reference to their results.

We thank the reviewer for pointing out this mistake, we have now changed this sentence to read "More recently, studies at the CERN cloud chamber have also shown that reactive iodine emissions can react to form iodic acid (Finkenzeller et al., 2023), which is the major iodine species driving both nucleation and growth of iodine oxoacid particles in pristine regions. The efficacy of iodine oxoacids to form new particles has been shown to exceed that of the $H_2SO_4 \cdot NH_3$ system at the same acid concentrations (He et al., 2021)."

> Line 58: please convert ng m−3 to pmol mol −1 which is frequently used in this study. Also line 58: please provide the measured range of the Koenig et al. (2020).

We have change this to read "0.013-4.62 pmol $mol^{-1}$"

> Lines 133-134: I do not understand how masses 95 and 97 can affect the bromide measurement? Is this instrument specific problem? Please specify.

Particles with large signals at at m/z 95 and 97 may contain MSA. Particles containing MSA will also contribute a large peak at m/z 79, which would give us a false positive for bromine identification. We have rephrased that sentence to read "Particles that contain methanesulfonic acid (MSA) have large A95 and A97 peaks, but will also have large a large A79 peak from MSA fragmentation; thus MA79 excludes negative and positive spectra where A79 was less than 5x A95 and 5x A97 to omit false positives from MSA."

> From the section 2.2, it seems the systematic errors of the reported bromine and iodine concentrations are more likely the lower limit than higher limit. It would be great if such systematic error can be reflected in the abstract and conclusions where such numbers are quoted.

Unfortunately, this is the statistical error for only the mass fractions within a population of particles, and not for the number fractions or mixing ratios quoted in the abstract and conclusions. We did, however, add this sentence to the conclusions "Normalized root-mean-square error for these measurements was approximately 20-60% for most of the atmosphere (Figure A3), and systematic biases due to unknown iodine speciation is likely below a factor of two (Section 2.2).

Lines: 254-255: This is interesting. However, the explanation is not convincing enough since gaseous and particulate bromine and iodine are constantly exchanging through heterogenous processes. Therefore, this cannot be a single way of accumulation of mass. It is also likely that the dry air prohibits such exchange, therefore resulting in net accumulation of bromide and iodine in aerosols.

Lines 337-344: following the same as the last comment, could the change in RH also a possibility?

Lines 351-353: going into the same line - difficult for me to imagine how RH is going to prevent condensable iodine species to condense onto particles, as laboratory experiments have pointed to a kinetic condensation of species such as $HIO_3$, which is the dominant mass contributor in at least the MBL1. However, the reverse may be more true: high RH may promote the release of iodine from particles, thereby reducing the iodine particle mass concentration.

Lines365-366: another line of evidence supports potential RH effect.

This string of comments relates to a major comment the reviewer makes above–namely, that we spend little time talking about heterogeneous recycling of iodine in aerosol. While it is true that we know that iodide and iodate may react heterogeneously with oxidants to liberate $I_2$, it is unclear from the literature if similar reactions exist for organically bound iodine. Since PALMS did not determine what form the iodine is in, we did not spend much time in our original submission speculating on this. The reviewer, however, is right to point out that it should be discussed in light of Section 3.2. Thus, we have added the following paragraph to Section 3.2:

"Figure 8 and A10 show that particles accumulate iodine as they age, but we do not suggest that the halogens are inert in aerosol after uptake. For example, it is known that iodide in aerosol can react heterogeneously with $O_3$ to form $I_2$, which is then liberated from the particles. This was traditionally thought to deplete all iodide in aerosol particles (Vogt et al., 1999), but field measurements consistently show non-neglible iodide in fine-mode marine aerosol (Gómez Martín 2022). Furthermore, iodate was traditionally thought to accumulate in particles, but several pathways to reduce aerosol iodate have been propose (e.g., Pechtl et al., 2007). Finally, organically bound iodine in aerosol makes up a significant fraction of the fine-mode iodine aerosol (Gómez Martín 2022), but its participation in heterogenous reactions that liberate iodine from aerosol is largely unexplored. Less is known about bromine liberation in nSSA. Unfortunately, in this study, PALMS did not determine what form the aerosol iodine or bromine is in. Thus, we are unable to quantitatively discuss the impact of heterogeneous recycling reactions in this work."

We have also made changes to the sentence in Lines 351-353:

"One possibility is that low $O_3$ or high water vapor concentrations are preventing

the formation of iodine or bromine aerosol or its precursors, directly affecting their partitioning from the gas phase to the aerosol phase Additionally, it is likely that dry particles are less efficient than wet particles at heterogeneously recycling aerosol iodine species (Sect. 3.1)."

And we also added the following sentence to Line 365:

"The stratosphere is an interesting foil to the MBL in that it is generally dry, and this may be further evidence that RH is important for aerosol iodine retention."

> Lines 367-383: meteoric aerosols presumably contain a lot of iron which promotes the conversion of iodide to molecular iodine2. This may be another reason for the strikingly lower aerosol iodine in meteoric aerosols than the sulfate-organic-nitrate / biomass burning aerosols.

We thank the reviewer for pointing out this reference. We have added that reference and the following text to Line 389: "One such proxy may be metal content, as meteoric particles are known to have smaller organic mass fractions (Murphy et al., 2021) and some meteoric metals like Fe can convert aerosol-bound iodine back to $I_2$ (Fudge et al., 1952)."

> Lines 384-393: along the same line, is it possible to check the correlation between metals such as iron vs. iodine content?

We have checked the correlation between A56 ($Fe^+$) and MA127 and did not find a significant correlation. Certainly, part of the issue is that most particles that are known to contain $Fe^+$ have little-to-no MA127 signal.

> Lines 412-413: inorganic emissions of what? Presumably halogens?

Noted. We have changed this to "inorganic emissions of halogens" and "organic emissions of halogens"

> Lines 459-460: The statement that water reduces iodine particle formation in the MBL is unfortunately incorrect. Experiments conducted under atmospheric conditions generally indicate a positive enhancement of particle nucleation rates with increasing water content. Specifically for iodine aerosol nucleation, recent evidence suggests that relative humidity above 2% has negligible impact on iodine aerosol nucleation in the marine boundary layer3. Please remove this statement. Furthermore, the high levels of iodine in nSSA are also likely attributable to missing heterogeneous chemistry processes.

We have added the following clause to Line 459: "... humid MBL, although this has been shown to be not the case for iodine oxoacid nucleation (Rorup et al., 2024)."

Lines 469-472: could the authors please specify what are the major iodine components in nSSA in GEOS-Chem for a comparison with the ATom measumements? Ideally it should be quantitative.

We thank the reviewer for their suggestion. We have added the following qualifiers to Line 469: "(irreversible uptake by HI and $I_2O_X$, X = 2,3,4)"

Lines 469-477 are difficult to follow, and more context is needed for general readers regarding what is consistent and what is not consistent between the conclusions of this study and Koenig et al. 20204. While reformulating the discussion, it is worth mentioning that the higher iodate in the LS would already indicates lower regeneration of iodine to the gas phase, thereby may lead to lower gas phase reactive iodine, assuming similar total nSSA iodine in the LS and UT.

We thank the reviewer for their comment. We have change this paragraph to now read "During the TORERO and CONTRAST campaigns, IO was reported to decrease from the tropical UT to the LS as aerosol iodine increased in a similar magnitude, suggesting that an additional source of iodine from the upper stratosphere is not needed to explain the increase in aerosol iodine (Koenig et al., 2020). Measurements from the CU HR-ToF-AMS during ATom show that aerosol iodine in the stratosphere was consistent with iodate, while most of the aerosol iodine found in the troposphere was iodide or organic iodine (Koenig et al., 2020). In GEOS-Chem 12.9.1, nSSA-iodine is formed through the same mechanisms in the stratosphere as it is in the troposphere (irreversible uptake by HI and $I_2O_X$, X = 2,3,4), which is inconsistent with the HR-ToF-AMS iodate results. Unfortunately, the PALMS data presented here did not confirm if the iodine LS aerosol was in fact iodate, but the aerosol iodine measurements between AMS and PALMS agrees to within 8% (median bias) for ATom-1 and ATom-2. The PALMS measurements also suggest that the iodine is organically bound, which is also not directly accounted for in the model. Thus, because low RH, iodate, and organically bound iodine would suppress heterogeneous chemical reactions that liberate iodine back to the gas phase, and because the model does not include these reactions and still underestimates aerosol iodine concentrations, a missing chemical pathway is more likely."

References:
(1) He, X.-C.; Tham, Y. J.; Dada, L.; Wang, M.; Finkenzeller, H.; Stolzenburg, D.; Iyer, S.; Simon, M.; Kürten, A.; Shen, J.; Rörup, B.; Rissanen, M.; Schobesberger, S.; Baalbaki, R.; Wang, D. S.; Koenig, T. K.; Jokinen, T.; Sarnela, N.; Beck, L. J.; Almeida, J.; Amanatidis, S.; Amorim, A.; Ataei, F.; Baccarini, A.; Bertozzi, B.; Bianchi, F.; Brilke, S.; Caudillo, L.; Chen, D.; Chiu, R.; Chu, B.; Dias, A.; Ding, A.; Dommen, J.; Duplissy, J.; El Haddad, I.; Gonzalez Carracedo, L.; Granzin, M.; Hansel, A.; Heinritzi,

M.; Hofbauer, V.; Junninen, H.; Kangasluoma, J.; Kemppainen, D.; Kim, C.; Kong, W.; Krechmer, J. E.; Kvashin, A.; Laitinen, T.; Lamkaddam, H.; Lee, C. P.; Lehtipalo, K.; Leiminger, M.; Li, Z.; Makhmutov, V.; Manninen, H. E.; Marie, G.; Marten, R.; Mathot, S.; Mauldin, R. L.; Mentler, B.; Möhler, O.; Müller, T.; Nie, W.; Onnela, A.; Petäjä, T.; Pfeifer, J.; Philippov, M.; Ranjithkumar, A.; Saiz-Lopez, A.; Salma, I.; Scholz, W.; Schuchmann, S.; Schulze, B.; Steiner, G.; Stozhkov, Y.; Tauber, C.; Tomé, A.; Thakur, R. C.; Väisänen, O.; Vazquez- Pufleau, M.; Wagner, A. C.; Wang, Y.; Weber, S. K.; Winkler, P. M.; Wu, Y.; Xiao, M.; Yan, C.; Ye, Q.; Ylisirniö, A.; Zauner-Wieczorek, M.; Zha, Q.; Zhou, P.; Flagan, R. C.; Curtius, J.; Baltensperger, U.; Kulmala, M.; Kerminen, V.-M.; Kurtén, T.; Donahue, N. M.; Volkamer, R.; Kirkby, J.; Worsnop, D. R.; Sipilä, M. Role of Iodine Oxoacids in Atmospheric Aerosol Nucleation. Science 2021, 371 (6529), 589–595. https://doi.org/10.1126/science.abe0298.

(2) Fudge, A. J.; Sykes, K. W. 25. The Reaction between Ferric and Iodide Ions. Part I. Kinetics and Mechanism. J. Chem. Soc. Resumed 1952, 119. https://doi.org/10.1039/jr9520000119.

(3) Rörup, B.; He, X.-C.; Shen, J.; Baalbaki, R.; Dada, L.; Sipilä, M.; Kirkby, J.; Kulmala, M.; Amorim, A.; Baccarini, A.; Bell, D. M.; Caudillo-Plath, L.; Duplissy, J.; Finkenzeller, H.; Kürten, A.; Lamkaddam, H.; Lee, C. P.; Makhmutov, V.; Manninen, H. E.; Marie, G.; Marten, R.; Mentler, B.; Onnela, A.; Philippov, M.; Scholz, C. W.; Simon, M.; Stolzenburg, D.; Tham, Y. J.; Tomé, A.; Wagner, A. C.; Wang, M.; Wang, D.; Wang, Y.; Weber, S. K.; Zauner-Wieczorek, M.; Baltensperger, U.; Curtius, J.; Donahue, N. M.; El Haddad, I.; Flagan, R. C.; Hansel, A.; Möhler, O.; Petäjä, T.; Volkamer, R.; Worsnop, D.; Lehtipalo, K. Temperature, Humidity, and Ionisation Effect of Iodine Oxoacid Nucleation. Environ. Sci. Atmospheres 2024, 10.1039.D4EA00013G. https://doi.org/10.1039/D4EA00013G.

(4) Koenig, T. K.; Baidar, S.; Campuzano-Jost, P.; Cuevas, C. A.; Dix, B.; Fernandez, R. P.; Guo, H.; Hall, S. R.; Kinnison, D.; Nault, B. A.; Ullmann, K.; Jimenez, J. L.; Saiz-Lopez, A.; Volkamer, R. Quantitative Detection of Iodine in the Stratosphere. Proc. Natl. Acad. Sci. 2020, 117 (4), 1860–1866. https://doi.org/10.1073/pnas.1916828117.